# The Protective Effects of Neurotrophins and MicroRNA in Diabetic Retinopathy, Nephropathy and Heart Failure via Regulating Endothelial Function

**DOI:** 10.3390/biom12081113

**Published:** 2022-08-12

**Authors:** Sergey Shityakov, Michiaki Nagai, Süleyman Ergün, Barbara M. Braunger, Carola Y. Förster

**Affiliations:** 1Division of Chemoinformatics, Infochemistry Scientific Center, Lomonosova Street 9, 191002 Saint-Petersburg, Russia; 2Department of Cardiology, Hiroshima City Asa Hospital, 2-1-1 Kabeminami, Aaskita-ku, Hiroshima 731-0293, Japan; 3Institute of Anatomy and Cell Biology, Julius-Maximilians-University Würzburg, 97070 Würzburg, Germany; 4Department of Anaesthesiology, Intensive Care, Emergency and Pain Medicine, Würzburg University, 97080 Würzburg, Germany

**Keywords:** diabetic retinopathy, diabetes mellitus, microvascular complications, diabetic nephropathy, heart failure, microRNA, neurotrophins

## Abstract

Diabetes mellitus is a common disease affecting more than 537 million adults worldwide. The microvascular complications that occur during the course of the disease are widespread and affect a variety of organ systems in the body. Diabetic retinopathy is one of the most common long-term complications, which include, amongst others, endothelial dysfunction, and thus, alterations in the blood-retinal barrier (BRB). This particularly restrictive physiological barrier is important for maintaining the neuroretina as a privileged site in the body by controlling the inflow and outflow of fluid, nutrients, metabolic end products, ions, and proteins. In addition, people with diabetic retinopathy (DR) have been shown to be at increased risk for systemic vascular complications, including subclinical and clinical stroke, coronary heart disease, heart failure, and nephropathy. DR is, therefore, considered an independent predictor of heart failure. In the present review, the effects of diabetes on the retina, heart, and kidneys are described. In addition, a putative common microRNA signature in diabetic retinopathy, nephropathy, and heart failure is discussed, which may be used in the future as a biomarker to better monitor disease progression. Finally, the use of miRNA, targeted neurotrophin delivery, and nanoparticles as novel therapeutic strategies is highlighted.

## 1. Introduction

To begin, miRNA signatures (miRNA) and characteristic neurotrophin levels in patient blood may potentially serve as blood-borne biomarker candidates.

Diabetes mellitus (DM) is a metabolic disease, characterised by unfavourably elevated blood glucose levels [1]. DM has several types, including type 1, type 2, maturity-onset diabetes, gestational diabetes, neonatal diabetes, and secondary causes due to metabolic syndrome, steroid use, etc. The main subtypes of DM are Type 1 diabetes mellitus (T1DM) and Type 2 diabetes mellitus (T2DM) [2]. T1DM is prevalent in children or adolescents, and is the result of an autoimmune reaction that destroys insulin-producing beta cells in the pancreas, finally resulting in absolute defective insulin secretion [3]. T2DM is caused by progressive loss of insulin secretion due to an insensitivity of the cells to extracellular insulin [3]. T2DM is believed to affect middle-aged and older adults subject to prolonged hyperglycemia resulting from poor lifestyle and dietary choices [1]. The pathogenesis for T1DM and T2DM is essentially different, and therefore, either case has its own etiology, presentation, and choice of treatments (reviewed in: [4]). Presently, lowering blood glucose levels is a fundamental part of the treatment regimen for diabetes. However, several observational studies fail to demonstrate a reduction in heart failure (HF) hospitalizations in diabetic patients treated with anti-hyperglycemic therapy [1]. On the other hand, in placebo-controlled trials, SGLT2i (sodium glucose transporter 2 inhibitors) reduced HF incidence [5]. In the meta-analysis, compared with placebo, HF risk showed a non-significant 10% reduction with the newer anti-hyperglycemic drugs (HR = 0.90, 0.80–1.01); use of DPP-4i and glucagon-like peptide-1-receptor-agonists GLP-1 RAs [6] was associated with nonsignificant modifications of the HF risk (+5% and −9%, respectively). The use of SGLT-2i was associated with a significant 31% reduction of the HF risk (HR = 0.69, 0.61–0.79, *p* < 0.001), with no heterogeneity (I^2^ = 0%, *p* = 0.741), suggesting a class effect. In type 2 diabetes mellitus (T2DM), SGLT-2i can reduce the risk of HF that is unrelated to improved glycemic control; DPP-4i and GLP-1 RAs behave as neutral [7]. Likewise, antioxidants have not yielded cardiovascular benefits in large clinical trials [8]. Another recent meta-analysis showed that pooled intention-to-treat analysis showed a reduced risk of stroke with SGLT2 inhibitors compared to DPP-4 inhibitors (hazard ratio HR, 0.89; 95% CI, 0.82–0.96; I^2^ = 25%; *p* = 0.25) and non-SGLT2 inhibitors (HR, 0.83; 95% CI, 0.77–0.91; I^2^ = 11%; *p* = 0.34). SGLT2 inhibitors were also associated with reduced cardiovascular outcomes and mortality risk in all comparisons [9].

Diabetic cardiomyopathy (DCM) is defined as ventricular dysfunction in the absence of hypertension, coronary artery disease, and valvular heart disease. In 1972, DCM was first reported from the findings of postmortem autopsy of four diabetic patients who manifested heart failure symptoms without coronary artery disease or valvular heart disease [10]. The American College of Cardiology Foundation, the American Heart Association [11], and the European Society of Cardiology in collaboration with the European Association for the Study of Diabetes [12] defined DCM as a clinical condition of ventricular dysfunction that occurs in the absence of coronary artery atherosclerosis and hypertension in patients with DM. Impaired coronary microvasculature is frequently observed in patients with T2DM, insulin resistance, and DCM [13,14].

Diabetes-related microvascular complications such as diabetic nephropathy constitute a significant public health problem. Diabetic nephropathy is a leading cause of chronic kidney disease and end-stage kidney disease [15]. Furthermore, most of the excess mortality risk observed in patients with diabetes may be related to the presence of diabetic nephropathy [16]. SGLT2i have recently emerged as a new class of oral glucose-lowering agents with pleiotropic effects including reduction in cardiovascular and kidney outcomes among patients with T2D [17]. In a recent meta-anlysis, SGLT2i slowed estimated glomerular filtration rate (eGFR) decline, lowered albuminuria progression, and improved adverse renal endpoints [18]. Nephropathy is an important microvascular complication of T2DM.

Interestingly, emerging evidence suggests that DR may share common pathophysiology with systemic vascular complications in DM. As a result, DR might reflect a widespread microcirculatory disease, not only in the eye, but also in vital organs elsewhere in the body [19]. Therefore, targeting immunometabolic pathways leading to diabetes-associated microvascular complications may strengthen the efficacy of preventive and therapeutic approaches. Moreover, the identification of common immunometabolic pathways for end organ damage in diabetes-related heart failure and diabetic retinopathy (DR) facilitates early diagnosis. Progression analysis and therapy development would tremendously improve the quality of life, and ultimately, the lifespan of affected patients [20,21]. These are to be carried out via analyses of patient blood.

miRNAs regulate gene expression at the posttranscriptional level and belong to a class of small, non-coding RNAs (~20–22 nucleotides long) [22,23]. Since the first miRNA was discovered in Caenorhabditis elegans in the 1990s, thousands of miRNAs have also been characterised in other species. High-throughput miRNA sequencing in whole blood [24] and plasma [25] assessed differential expression of numerous miRNAs associated with CVDs under physiological and pathological stress [26,27,28,29]. The rapid and cost-effective technologies of microarray- or PCR-based miRNA screening can be applied to detect circulating RNAs in various body fluids. However, non-annotated non-coding RNAs are not detected. In the context of cardiovascular disease (CVD), functional high-throughput screens have been applied to RNA-seq data. In addition, to identify miRNAs that regulate, for example, cardiomyocyte regeneration, a library based on human miRNA mimics, a high-content microscopy screen, and a functional high-throughput screen were used [30]. In summary, these studies are the first examples of the discovery of non-coding RNAs that may be of physiological relevance to CVDs using high-throughput functional screening.

On the other hand, some molecules, like neurotrophins responsible for cellular survival, development, and function, might be useful for treating diabetic complications and heart disease via regulation of endothelial function [31]. However, these proteins have not been widely recognized as promising therapeutic molecules since the first discovery of nerve growth factor (NGF) in the 1950s. One of the mechanisms by which neurotrophins affect endothelial cell functions is the stimulation of systemic mobilization of hematopoietic progenitors [32]. Furthermore, neurotrophins could also promote therapeutic neovascularization by stimulation of tropomyosin-kinase receptors in mouse ischemia models [33].

It is also well known that neurotrophins (BDNF) could protect neuronal tissue and improve CNS function in patients with diabetes via tyrosine kinase receptor B (TRKB) and cAMP-response element binding protein (CREB)-mediated pathways [34]. Overall, these investigations provide an insight into the therapeutic application of neurotrophins and the possibility for designing their derivatives, analogs, or formulations with specific pharmacological properties [35].

In this review, evidence is presented that the elucidation of common immunometabolic pathways to end-organ damage in diabetes-related heart failure, diabetic retinopathy (DR), and nephropathy, respectively, could greatly help to facilitate early diagnosis, progression analysis, and therapy development of all three conditions. Being a uniquely specific and non-invasive assessable measure of diabetic microvascular damage, analyses of DR-related early microvascular changes in the eye could help to detect microvascular-associated diseases such as diabetic nephropathy and DCM in otherwise asymptomatic patients. Novel treatment regimens including microRNA (miRNA)- and neurotrophin-based strategies are being presented.

## 2. Eyes, Renal and Cardiac Microvasculature—From a Microvascular Signature to End Organ Damage in Diabetic Retinopathy, Diabetic Nephropathy and Diabetic Cardiomyopathy

### 2.1. The Blood-Retinal Barrier

The retina is part of the central nervous system (CNS) [36], and therefore, shares considerable similarities with the brain. This could be demonstrated by characteristics such as the blood-retinal barrier (BRB), which shields retinal neurons from harmful, blood-borne substances comparable to the protection of brain neurons by the blood-brain barrier (BBB) [37,38,39]. The BRB tightly controls not only the flux of fluid and blood borne substances into and out of the retina, but also confers to obtain retinal immunoprivilege [40,41]. In primates and rodents, the inner retina obtains its blood supply from blood vessels dividing the central retinal artery to eventually form three capillary plexuses. These are located in the nerve fiber layer and in the inner and outer plexiform layers of the retina (Figure 1) [42]. Photoreceptors, however, are nourished by choroidal vessels that finally form a very dense capillary network. The choriocapillaris is adjacent to the extracellular matrix of the Bruch‘s membrane and the retinal pigment epithelium (RPE) [42] (Figure 1). Given the fact that the retina is supplied by two different vascular networks, two BRBs, the inner (iBRB) and outer BRB (oBRB), are also formed [43].

The outer BRB (oBRB) is formed by tight junctions between RPE cells (Figure 1) to protect the retinal neurons from blood-borne substances that might otherwise easily access the retina. This is explained by the endothelium of the adjacent choriocapillaris being fenestrated, and therefore, being highly permeable [37,44].

The inner BRB (iBRB) is formed by tight junctions between endothelial cells (Figure 1) of the retinal capillaries and is essentially analogous to the blood-brain barrier (BBB) [45]. It is exceptionally tight and restrictive and also serves as a physiologic barrier controlling ion, protein, and water flux into and out of the retina [46].

Histologically, the iBRB is composed of endothelial cells sealed by a well-developed apical junctional complex between neighbouring endothelial cells. These restrict paracellular flux. As with endothelial cells of the BBB [47], they have few vesicles and no fenestrae formation reducing transcellular permeability [48]. The cells also express characteristic transporter patterns, amongst them, efflux transporters of the multi-drug resistance family [48,49]. As in the brain, formation of this very tight phenotype requires interaction with tissue-specific neighbouring cells like neurons, glia, and pericytes. Ultimately, this leads to the concept of the neurovascular unit (Figure 1) [48]. The glia and pericytes provide endothelial support and maintenance, and as for the BBB, their damage inevitably results in reduced endothelial tightness [50]. Recent studies have described the relative contribution of signaling pathways from glia as essential for the development and maintenance of the iBRB [19,24,27]. For more in-depth insight, please refer to the clinical literature, e.g., [43,51].

In particular, endothelial cells and pericytes have a close relationship. Not only do they share a common basement membrane, but molecular communication through signaling molecules such as TGFβ has been shown to be essential for their regular development and maintenance [52,53,54,55]. Accordingly, it has recently been shown by Braunger and colleagues that deletion of TGFβ signaling in the entire eye in mice affects, amongst other factors, pericytes and endothelial cells. It results in a phenotype that largely reflects that of humans suffering from diabetic retinopathy [54].

### 2.2. The iBRB and Diabetic Retinopathy

Diabetic retinopathy (DR) is a leading cause of blindness in the working population of the western world [56]. Approximately 60% of patients diagnosed with type 2 diabetes and almost all patients diagnosed with type 1 diabetes demonstrate certain symptoms of DR [57]. DR is positively correlated with metabolic stress as induced by hyperglycaemia [58]. It thereby results in an increase of reactive oxygen species (ROS) and activation of the advanced glycation end products (AGEs) pathway and the local renin-angiotensin system (RAS), promoting cellular dysfunction and apoptosis (Figure 2 and Figure 3) [59,60].

DR is clinically classified based on the changes observed in the retinal microvasculature. Therefore, DR is subdivided in a either proliferative or non-proliferative form, respectively [57]. In early, non-proliferative DR, the reduction of pericyte numbers, termed as pericyte drop-out, and hyperglycemia-driven vascular basement membrane thickening (due to the excess accumulation of basement membrane components) occur as initial morphological changes [52,62,63,64].

This results in a dysfunction of the neurovascular unit, and thus, further promotes microvascular changes such as microaneurysms, microhemorrhages, nerve fiber infarcts (termed as cotton-wool spots), exudate deposits (termed as hard exudates), acellular and non-perfused capillaries, and venous caliber changes [52,65]. These microvascular alterations eventually result in decreased integrity and increased permeability of the retinal vessels. However, these described alterations are generally asymptomatic, even at advanced stages [66]. With progression of the disease, occlusion of retinal vessels leads to ischemia and ischemic infarcts of the nerve fiber layer [65]. Subsequently, hypoxia-associated, pro-angiogenic factors, such as vascular endothelial growth factor (VEGF), become upregulated. This is accompanied by pro-inflammatory factors, resulting in neuroinflammation, an increased vascular permeability, and retinal/vitreal neovascularization, a scenario clinically named as proliferative DR [43,65,67,68,69]. These neovascularizations are fragile and mostly consist of fenestrated endothelium, thus making them more prone to leakage and hemorrhaging [65,69]. In the late stage of the disease, repeated vitreous hemorrhages lead to the formation of fibrous, tractive vitreal membranes, resulting in retinal detachment, and thereby, pronounced vision loss [65,69].

Moreover, diabetes-associated macula edema (Figure 4), which affects central vision due to pathological accumulation of extracellular fluid in the macula region, can develop at any stage of the disease. However, it occurs more frequently as the severity of DR increases [70].

However, studies in cell cultures, animal models and human samples show clear evidence that not only the retinal vessels, but rather the retinal neurons, glia cells, the choroid, and RPE, are also affected by the metabolic and signaling challenges of diabetes [52,71,72,73,74,75]. Accordingly, retinal diabetic neuropathy is characterized by a degeneration of the inner retinal neurons and reduced neuronal function, even before DR becomes clinically symptomatic, e.g., preceding microvascular changes [76,77]. Experimental animals with diabetes show ultrastructural changes and a loss of tight junction integrity in the RPE [78,79]. Accordingly, an increased leakage of blood content can be observed in human diabetic patients and in diabetic animals indicating an alteration of the oBRB in DR [78]. Moreover, in particular, the observed neuroinflammation is not only a consequence of barrier dysfunction. It is also the result of an early local mechanism contributing to barrier alteration and leukostasis [19,56,57]. Accordingly, activation of microglia and Müller cells concomitant with secretion of inflammatory mediators was reported as an early observation in DR [60,80]. An increased expression of pro-inflammatory cytokines such as tumor necrosis factor (TNF)-α, the interleukins IL-1β, IL-6, IL-8, and CC-chemokine ligand 2 (CCL2), have been determined in human ocular samples, and could, moreover, be positively correlated with DR severity [43,57,81,82,83,84,85,86,87]. Moreover, exposure of endothelial cells to these pro-inflammatory cytokines leads to an increased expression of certain cell adhesion molecules (ICAM-1, VCAM-1 and E-selectin). They are suitable for the mediation of leukocyte docking and trespassing of the endothelial barrier [88]. Additionally of note is the fact that antioxidant application was reported to reduce levels of ICAM-1 and endothelial cell apoptosis in vitro [89]. Finally, the well-established potency of corticosteroid-based therapy for patients suffering from diabetes-associated macular edema is clearly a convincing argument for an involvement of inflammation in DR [19,67,68]. There is evidence that the benefits of corticosteroid treatment are not limited to macular edema, but may also slow the overall progression of DR [43,90,91,92].

Intriguingly, as the retinal vessels can easily be visualized through funduscopy and/or angiography [31,72,73], microvascular alterations caused by systemic diseases diabetes can easily be observed in retinal blood vessels and can allow for predictions regarding the vascular status of other organs, e.g., the heart, the kidney, and/\=or the brain.

### 2.3. Heart Failure in Diabetes Mellitus

Recent results suggest that DR and diabetes-associated heart failure might have a similar pathophysiological, inflammatory, microvascular background [93,94]. More diabetic retinopathy-specific data was observed in the publication from the Atherosclerosis Risk In Communities (ARIC) study where diabetic retinopathy was shown to be associated with an excess risk of developing HF [94]. Data from the Multiethnic Study of Atherosclerosis (MESA) also showed that microvascular complications were associated with more concentric hypertrophy on echocardiography, a hallmark of heart failure preserved ejection fraction (HFpEF) [95]. In the PROMIS-HFpEF study, a high prevalence of coronary microvascular dysfunction in HFpEF was shown in the absence of unrevascularized macrovascular coronary artery disease [96] (Shah, et al., 2018).

DR is the most common complication of diabetes. In addition, DR patients possess an undue risk for systemic vascular complications such as stroke, coronary artery disease, heart failure, and nephropathy. As a uniquely specific and non-invasive measure of diabetic microvascular damage, analysis of DR-related early microvascular changes in the eye and of blood-borne markers might be very suitable as a predictor of inflammatory microvascular diseases. These can affect the heart in otherwise asymptomatic patients:

The diabetic heart is characterized by metabolic disorders, often being associated with local inflammation, oxidative stress, myocardial fibrosis, and cardiomyocyte apoptosis.

DCM is not uncommon and is associated with increased morbidity and mortality in patients with diabetes [97]. Microvascular abnormalities are also considered to be involved in altering cardiac structure and function. Three major defects, such as endothelial dysfunction, alteration in the production/release of hormones, and shifts in the metabolism of smooth muscle cells, have been suggested to produce damage to the small arteries and capillaries (microangiopathy). This is due to hyperglycemia and the promotion of the development of the DCM Role of microangiopathy in diabetic cardiomyopathy [14]. Other pathogenetic mechanisms involved in decreasing myocardial contractility in DCM are impaired calcium homeostasis, renin–angiotensin system upregulation, increased oxidative stress, altered substrate metabolism, and mitochondrial dysfunction [98]

As pointed out, recent results suggest that DR and diabetes-associated heart failure have a similar pathophysiological, inflammatory, microvascular background. The occurrence of DR is, therefore, an independent predictor of heart failure. The identification of selected biomarkers integrating these processes is of great importance in order to detect, prevent, or even treat diabetes-associated heart failure at an early stage once it has been established [99].

The knowledge gained could also help elucidate general molecular principles of inflammation-based heart failure. In that context, the development and use of combination therapy with anti-VEGF and selected anti-inflammatory agents like corticosteroids with other previously available treatments, such as panretinal photocoagulation. Early trials showed promising results, panretinal photocoagulation is under investigation as a reasonable clinical strategy. This is to reduce the burden of hyperclycemia and intravitreal injection-induced inflammation in diabetic eye disease [100].

Of special interest is common immunometabolism pathways to end organ damage in diabetes-related heart failure and diabetic retinopathy (DR) to facilitate early diagnosis, progression analysis, and therapy development. These are to be carried out via biochemical testing on the blood basis of early pathological changes in the development of DR during routine eye examinations in diabetics.

### 2.4. Diabetic Nephropathy

Nephropathy is a common complication affecting almost one-third of patients with diabetes [101]. Both microalbuminuria and proteinuria are markers of increased cardiovascular risk [102,103]. In type 1 diabetes, the relative risk of DCM is 1.2-fold in patients with microalbuminuria [104] and 10-fold in patients with proteinuria [105] compared with patients with normoalbuminuria. In type 2 diabetes, the risk of DCM is increased 2- to 3-fold in microalbuminuria [106] and 9-fold in proteinuria [107]. The onset of microalbuminuria is associated with an excess of cardiovascular risk factors, including hyperglycemia, dyslipidemia, and numerous prothrombotic and atherogenic changes [108,109]. Suggested mechanisms linking renal and cardiovascular disease include endothelial dysfunction, abnormalities of the renin–angiotensin system, and widespread vascular basement membrane defects [97].

The progression of DR and diabetic nephropathy for patients with T2DM was discordant. The Renal Insufficiency and Cardiovascular Events study found that the progression of diabetic nephropathy did not affect 41.4% of T2DM patients with advanced DR [110]. Glycemic variability over the long term was found not to affect the progression of DR but could predict the presence of DN [111]. These results could be explained by the pathogenesis of DR and diabetic nephropathy, which could be affected by different risk factors. Moreover, the previous meta-analysis has already found that DR should be regarded as a useful status for diagnosing and predicting diabetic nephropathy for patients with T2DM [112]. Another recent meta-analysis found significant associations between DR and subsequent DN risk for patients with T2DM [113].

Taken together, the lack of early biomarkers, coupled with the fact that the earlier stages of diabetic cardiomyopathy and nephropathy are mostly asymptomatic, makes detection a clinical challenge. Considering future research, the comparative study of the molecular effects of diabetic sequelae on the integrity of microvasculature of eyes, hearts, and kidneys might lead to a better understanding of the common mechanisms of diabetic microvascular damage. More specifically, it could elucidate the time of occurrence of microvascular diseases leading to damage to the end organs. In the following section, modalities to elucidate prevailing immunometabolic pathways to end-organ damage in diabetes-related heart failure, nephropathy, and retinopathy (DR), respectively, are being presented, which might be appropriate readouts to help facilitate early diagnosis, progression analysis, and therapy development for all three conditions. 

As potential blood-borne candidates for DR, the microRNA (miRNA) signatures specifically altered in HFpEF and HFrEF in DM patient blood are proposed for monitoring [93]. Additionally, the monitoring of characteristic neurotrophin levels in the blood of those who experienced pathological changes in the development of DR during routine examination of the eye in diabetics are being proposed [114,115,116,117,118].

## 3. Current Treatment Options

Treatment options are missing in non-proliferative DR; therapy is mostly limited to pan-retinal photocoagulation [51]. While this treatment reduces the risk of visual loss and blindness, it is more preserving in nature rather than correcting visual acuity. Pan-retinal photocoagulation is furthermore reportedly associated with deterioration in visual function, potentially affecting quality of life with respect to day-to-day living or a person’s ability to drive [51]. Furthermore, pan-retinal photocoagulation does not prevent associated active disease, such as vitreous haemorrhages. Therefore, this form of treatment is generally postponed for as long as possible [51].

Early trials raise confidence for intravitreal anti-vascular endothelial growth factor (VEGF) injections in severe non-proliferative DR and proliferative DR, respectively [119]. Some major concerns regarding this potential treatment are that intravitreal injections are invasive, prone to infection and inflammation, and are perceived as inconvenient. Alternative treatments for DR that may help to prevent progression to sight-threatening disease are, therefore, urgently needed. In particular, treatments could be conducted outside of a clinical environment, such as oral or topical formulations.

## 4. Future Perspectives

### 4.1. Targeting Inflammatory Mechanisms in Diabetes Mellitus

Various studies delivered compelling evidence for the possible link between inflammation and diabetes in animal models, revealing the key role of TNF-alpha in this pathology [120,121]. Some biomarkers and mediators, such as C-reactive protein, IL-6, fibrinogen, plasminogen activator inhibitor-1, and sialic acid, were determined in elevated concentrations based on epidemiological data, connecting diabetes and inflammation [122]. However, the etiology of T1D is mainly an autoimmune inflammation of insulin-producing pancreatic beta cells, while T2D is a metabolic disorder [123,124]. Therefore, given cytokines’ (IL-1beta) and TNF-alpha involvement in the process of beta-cell pathology, they might be plausible therapeutic targets for T1D treatment [121]. On the other hand, due to the B-cells’ immunity contribution to the adipose tissue inflammation as a result of the T2D metabolic imbalance, some B-cell knockout mice models and anti-CD20 therapies were proposed to improve metabolic and inflammatory phenotypes [125]. DR progresses through different pathophysiological pathways, namely in the form of oxidative stress, inflammation, and stimulation of the growth factor in the eye’s vasculature [126]. Increasing evidence has been given that inflammatory mechanisms are chiefly important in the pathogenesis of DR [86,127]. In detail, the upregulation of cytokines such as IL-6, IL-1β, IL-8, TNF-α, ICAM-1, vascular cell adhesion molecule V-CAM, selected integrins, and other proinflammatory cytokines and cell adhesion receptors have been demonstrated to lead to persistent low-grade inflammation. It is hypothesised that these actively play a part in the development of DR-associated damage to the iBRB, inducing breakdown, leading to subsequent macular edema formation, and promoting retinal neovascularization [127]. 

Moreover, VEGF and angiopoietins have shown themselves to be critical in the pathogenesis of diabetic eye disease. Worth noting is a novel biomarker specific for the development of proliferative DR, elevated NLRP3 inflammasome levels in the vitreous have been shown to correlate positively with the onset and progress of this variant of the disease [128].

Regarding adipose tissue inflammation and heart failure, several clinical studies have shown an association between systemic insulin resistance, inflammation and heart failure. For example, glucose metabolism is impaired in patients without type 2 diabetes who have idiopathic dilated cardiomyopathy [129].

Diabetic nephropathy has, so far, not been traditionally considered an inflammatory disease. However, recent studies have shown that kidney inflammation is crucial in promoting the development and progression of diabetic nephropathy [130].

Following this notion, currently, emerging therapeuties targeting inflammation are under evaluation, including the blockade of angiopoietin 2 and other inflammatory targets such as interleukins IL-6, IL-1β, plasma kallikrein, and integrins [126].

### 4.2. Identification of a Common MicroRNA Signature in Damage in Diabetic Retinopathy, Diabetic Nephropathy and Heart Failure

Diabetes mellitus is a metabolic disease, with miRNAs playing a crucial role in regulating metabolism [131].

miRNAs are highly conserved, endogenous, small, non-coding RNAs with a length of ~22 nucleotides, that regulate gene expression by binding to partially complementary sequences of mRNA [132]. In cardiac insufficiency, chronic immune activation and aberrant miRNA expression are consistently evident [133,134].

On the other hand, blood, serum, and plasma are accessible data sources for physiological and pathological status. Recently, it was shown that miRNAs in a more stable form circulate in the blood. This would suggest that circulating miRNAs are used as biomarkers for cardiovascular diseases such as DR and diabetes-associated heart failure. In addition, it has been shown that restoring normal levels of altered miRNAs is a promising approach to prevent diabetes-related changes. The assessment of blood-borne DR microRNA profiles might, therefore, present clinicians a unique opportunity to detect the onset of diabetic microvascular damage at an early stage, thus allowing for the performance of a blood-borne marker-based progression analysis.

New approaches to treat DR are being developed via precision medicine that aims to target novel epigenetic changes associated with the development of DR: amongst these approaches, targeting specific miRNA is a very promising avenue for managing DR [134,135]. The restoration of altered miRNA expression to normal levels has recently been shown as a promising approach to prevent diabetes-associated changes [2,7,20].

On one hand, it is known now that several miRNAs have been associated with the severity of DR, and future studies could even pinpoint whether circulatory miRNAs could serve as novel biomarkers to detect or predict DR [134]. On the other hand, current functional screenings have identified miRNA as multi-cellular regulators of heart failure [136]; among the differentially regulated entities, miR-155 plays an important role in the inflammation in the heart in diabetes-associated heart failure; these regulate processes such as hypertrophy, fibrosis (cardiac fibroid collagen content), and inflammation [137]. miR-155, therefore, plays an important role in the immune system in mammals as well as in HF, and is abundantly expressed in T cells, B cells, and monocytes [138,139,140,141]. An STZ-induced diabetic heart expresses higher levels of miR-195, and silencing miR-195 reduces DCM [142]. MiR-141 is also increased in the diabetic heart and influences mitochondrial function and ATP production [143]. Palmitate-stimulated neonatal rat cardiomyocytes (NRCs) and diet-induced obese (DIO) mouse heart also showed increased expression of miR-451, which lowers LKB1/AMPK signal transduction [144]. The expression of miR-133a decreases the Glut4 expression with the consequence of a decrease of the insulin-mediated glucose uptake in NRCs [145]. Remarkably, the diabetic heart is also characterized by a clear downregulation of recently discovered cardioprotective miRNA [146]. The development of further methods and research to establish a greater understanding of miRNA alterations in T2DM related to HF might become an important future focus area. Specifically, to assess more profoundly the association of these changes with respect to HfpEF or HfrEF, there will be greater importance placed on the ability to relate DR to HF subtype. This approach could facilitate the development of concepts for early detection and progression analysis of heart failure in diabetic patients. The relative expression of heart failure-related miRNA in cases of DR has, however, not been investigated so far.

miR-155 negatively regulates the expression of target gene E26 transformation-specific Sequence 1 (ETS-1) and its downstream factors VCAM-1, MCP-1, and cleaved caspase-3, thus mediating the inflammatory response and apoptosis of human renal glomerular endothelial cells [147]. The miR-155 expression was increased in renal tissues of DN patients, and mainly expressed in glomerular vascular endothelial cells, mesangial cells and renal tubule interstitium [148]. The results of luciferase reporter gene showed that ETS-1 may be a potential target gene of miR-155. Further detection of serum miRNAs in diabetic patients showed abnormal expression of miR-155 in diabetic patients compared with healthy controls, and the expression of miR-155 was significantly different in microproteinuria and macroproteinuria groups, and was positively correlated with eGFR in diabetic nephropathy patients and negatively correlated with urinary protein excretion rate. Triptolide upregulated BDNF by inhibiting miR-155-5p, thus inhibiting oxidative stress and inflammatory damage and alleviating podocyte injury in diabetic nephropathy mice [149]. miRNA-195 promotes apoptosis of podocytes under high-glucose conditions via enhanced caspase cascades for BCL2 insufficiency [150]. The abated miRNA-195 expression protected mesangial cells from apoptosis, suggesting that the antiapoptosis in a miRNA-regulated manner may play an important role in the early stages of diabetic nephropathy [151] (Chen et al., 2012). miR-455-3p suppresses renal fibrosis through repression of ROCK2 expression in diabetic nephropathy [152]. In the DN mouse model, Circ_0000491 knockdown inhibited high glucose-induced apoptosis, inflammation, oxidative stress, and fibrosis in SV40-MES13 cells by regulating miR-455-3p/Hmgb1 axis [153].

In T2DM retinopathy, miR-155 plays an important role in the pathogenesis of T2DM retinopathy by regulating the Treg cells with TGF-β. rs767649 polymorphism in the pre-MIR155 gene is associated with DR in T2DM, and miR-155 plasma levels might be associated with T2DM [154]. miR-155-5p expression was significantly upregulated in human retinal microvascular endothelial cells induced by high glucose. After inhibiting the expression of miR-155-5p, cell proliferation, angiogenesis, and VEGF protein levels were significantly downregulated, whereas miR-155-5p mimics had the opposite effect. miR-155-5p is closely associated with diabetic macular edema and is a potential target for refractory diabetic macular edema treatment [155]. miR-195 regulates SIRT1-mediated tissue damage in DR [156]. miR-195 knockdown led to the downregulation of the mRNA and protein expression levels of BAX and the upregulation of the mRNA and protein expression levels of SIRT1 and BCL-2 as well as improvement in cell growth and a decrease of the apoptosis rate [157]. miR-195 is overexpressed in DR, and its targeted relationship with SIRT1 inhibits the growth of cells in the retina and accelerates apoptosis [157]. miR-455-5p ameliorates high glucose-induced apoptosis, oxidative stress and inflammatory via targeting SOCS3 in retinal pigment epithelial cells [158].

Another promising biomarker to diagnose various cardiomyopathies, including heart failure and diabetic cardiomyopathy, might be miR-21 [159]. This 22-nucleotide long microRNA has a paradoxical effect on cardiomyocytes against stress-overloaded conditions. It activates fibroblast to trigger the fibrosis process, stimulating the proliferation of the heart cells at the same time. Besides, the high level of miR-21 expression might be linked to the suppression of programmed cell death protein 4 in head and neck squamous cell carcinoma [160].

One prospective long-term goal could, therefore, be the development of miRNA therapeutics for the prevention or treatment of diabetes-associated, inflammation-based heart failure: by targeting specific miRNAs, which govern whole pathways implicated in the pathogenesis of both DR and diabetes associated heart failure, it could be possible to develop novel diagnostic and therapeutic avenues to tackle the life-threatening condition of diabetes-associated HF.

Such therapeutic strategies could be developed utilizing miRNA therapeutics (Figure 5 and Figure 6). Establishing greater evidence and understanding has, moreover, even demonstrated that noncoding RNAs (ncRNAs) may also be expressed in various ways. DR expression may play a vital role in the development of DR. Amongst the non-coding RNAs, ncRNAs, besides miRNAs, and also the long ncRNAs (lncRNAs), have recently been described for their regulatory functions [135].

For future treatment regimens, several miRNA delivery strategies have been developed for use in practice, such as viral vectors, plasmid, piggybacks expression vectors, nanoparticles, exosomes, and liposomes [134].

However, although plenty of miRNA-based therapeutics have been investigated in preclinical studies, only a few of these have been selected for further clinical development. Challenges concern difficulties in proper target selection, varying stability in body fluids, and insufficient target specificity. It also relates to off-target effects, which remain to be resolved in the future, to optimize the delivery and efficiency of miRNA-based therapeutics. As miRNAs have been shown to be important in basic potential, there could be greater expectations from miRNA-based constructs.

#### 4.2.1. Neurotrophin Signaling and Neurotrophin-Related Treatment Opportunities in Diabetic Retinopathy

Also of significance, several components of the neurotrophin family are dysregulated upon diabetic challenges [161,162,163,164]. The mammalian neurotrophin family of growth factors comprises four ligands: nerve growth factor (NGF) [165], neurotrophin-3 (NT-3) [166], neurotrophin-4/5 (NT-4/5) [167], and brain-derived growth factor (BDNF) [168]. Initially, neurotrophins are expressed as precursor proteins (pro-neurotrophins) and proteolytically cleaved to their mature forms. This is to signal through tropomyosin receptor kinases (Trks) A-C and neurotrophin receptor p75 (p75^NTR^), even though biological activity has also been attributed to pro-neurotrophins [33,169,170,171,172].

While each neurotrophin has altered binding affinities to these receptors, it is well-established knowledge that the activation of Trk receptors primarily mediates cell survival, whereas p75^NTR^-mediated signaling promotes, with some exceptions, degeneration of neuronal and non-neuronal cells [94,129,134].

In streptozotocin (STZ)-induced diabetic rats, p75^NTR^ and proNGF are upregulated at an early stage in glial cells and pericytes [161,162,163,164], concomitant with an impaired TrkA receptor phosphorylation that shifts the TrkA/p75^NTR^ balance towards activation of the pro-apoptotic p75^NTR^ receptor in retinal ganglion cells. It eventually results in neuronal cell death [162]. Accordingly, in human ocular samples (aqueous humor, vitreous, and retinae) from diabetic patients and in STZ-induced diabetic retinae, an accumulation of proNGF and reduced NGF levels were observed [161]. Upregulation of p75^NTR^ and proNGF was furthermore associated with an increase of tumor necrosis factor alpha (TNFα) and alpha-2-macroglobulin (α_2_M) in the retina, pointing again towards a proinflammatory and neurotoxic environment [163]. Intriguingly, therapeutic treatment of diabetic rats with vitreal injections of a p75^NTR^ small-molecule antagonist (THX-B) or an anti-(pro)NGF blocking mAb (NGF30 mAb [173]) significantly reduced retinal TNFα and α_2_M levels and prevented death of retinal ganglion cells [163]. Accordingly, STZ-induced diabetic p75^NTR^ knockout mice showed an attenuation of the diabetes-induced increases in proNGF, nuclear factor κB (NFκB), phospho-NFκB, and TNF-α in the retina, concomitant with a reduced decrease of retinal NGF expression and mitigated retinal ganglion cell loss [174]. STZ-induced diabetic p75^NTR^ knockout mice were further protected against diabetes-induced BRB breakdown [174]. Similarly, STZ-induced diabetic rats treated with NGF eye drops showed a not significant trend toward protection of retinal ganglion cells from diabetes-induced degeneration [175]. Taken together, this data indicated that the proNGF–p75^NTR^ axis contributes to retinal inflammation and vascular dysfunction in STZ-induced diabetic retinae [174].

However, although current knowledge of neurotrophin signaling in diabetic retinopathy predominantly centers around the proNGF/NGF axis, other neurotrophins, such as BDNF, NT3, and NT4/5 are also dysregulated in diabetic retinae [176,177,178]. There are studies reporting significantly increased neurotrophin levels (BDNF, NGF, NT3, and NT4/5) in the vitreous of diabetic patients and in animals with experimentally induced proliferative diabetic retinopathy [177,178], but others demonstrate significantly decreased BDNF and NGF levels in serum and aqueous humor of diabetic patients [179,180]. These reports, contradictory at first sight, may well be attributed to the fact that investigation of diabetic retinal tissue most likely originated from diabetic patients in the late stages of the disease. Meanwhile, other samples (vitreous/aqueous humor) may have been collected during surgery at earlier stages of the disease.

Studies of the molecular interactions of BDNF and NGF in the diabetic retina have shown that both molecules exhibit pro-angiogenic effects and contribute to the formation of neovascularization in proliferative diabetic retinopathy [176]. This is either directly, by binding to Trk receptors on endothelial cells, or indirectly, by promoting vascular endothelial growth factor (VEGF) expression in other cells [177,181,182,183,184,185,186]. Thus, inhibition of VEGF by intravitreal injection of bevacizumab, a humanized anti-VEGF monoclonal antibody commonly used to inhibit neovascularization in patients with proliferative diabetic retinopathy, results in marked reduction of retinal NGF levels [187].

Notably, comparable to patients with cardiovascular disease, decreased serum BDNF and NGF levels can be detected before the onset of clinically manifested diabetic retinopathy, and are, therefore, discussed as biomarkers and critical indicators for the development of diabetic retinopathy [186]. 

#### 4.2.2. Targeted Delivery of Neurotrophic Factors to the Retina

A novel approach for DR treatment might be via API delivery to the retina using various drug delivery vectors, including various organic and inorganic nanoparticles (NPs), proteins, adenoassociated virus vectors (AVV), hydrogels, and even cellar implants to prevent retinal damage [188,189,190]. The use of NT-conjugated magnetic nanoparticles (MNTs) has been shown to be a strategy with potential to prevent the loss of retinal ganglion cells induced by oxidative stress damage [188]. Indeed, these MNTs tend to progressively migrate from the vitreous chamber to the retina within 24 h after eye microinjection, maintaining the NGF, GDNF, and FGF-2 factors in situ [188]. On the other hand, stable and biocompatible NPs incorporating CNTF and OSM were recently formulated and applied for the treatment of retinitis pigmentosa. The results of this study have shown significant photoreceptor preservation in an in vitro mouse model, offering long-term efficacy [191,192]. Direct intravitreal injection of CNTF, PEDF, and FGF2 can also be performed [193] to access photoreceptor survival and macro/microglial reactivity in two rat models of inherited retinal degeneration [194]. As an outcome, these neurotrophic factors were able to improve the outer segment morphology of photoreceptors and maintain their quantity [195].

Similarly, the subretinal delivery of AVV-CNTF was also measured in adult Long–Evans rats to improve electroretinogram (ERG) amplitudes caused by the retinal degeneration [193]. Despite some promising results, a few dose-related side effects were detected, such as a change in rod photoreceptor nucleus phenotype and a paradoxical decrease of ERG amplitudes, most likely due to the changes in gene expression [196].

Furthermore, Bush and colleagues showed that some neuroprotective factors can be successfully delivered to the retina by cells transfected with the human CNTF and sequestered within capsules suitable for intraocular implantation [197]. This Phase I clinical trial indicated that CNTF using encapsulated cell implants is safe for the human retina, even with severely compromised photoreceptors.

On the other hand, the usage of thermosensitive hydrogel enriched by CNTF for retinal ganglion cells (RGC) protection was proposed by Lin and co-authors. In this study, a pre-hydrogel liquid containing chitosan, hydrophobic macrolide immunosuppressant, CNTF, and the gelling agent, was directly smeared on the injured site, exhibiting in vivo RGCs protective action against the adverse effects caused by traumatic optic nerve injury [198].

Other alternative strategies can be proposed to regulate NT levels in the retina indirectly by implementing amphiphilic cyclodextrins (CDs). These molecules have already been shown in different studies to be promising drug delivery vectors comprising native and modified CDs [199,200,201]. In one study, neurotrophic signaling was restored by the CD-mediated exogenous cholesterol delivery using methyl-β cyclodextrin via de novo lipogenesis in the NT-dependent cell survival [202]. It could also be achieved by stabilizing retinal membrane configuration and lipid composition to prevent cell death in many forms of retinopathy [202]. Despite the limitation of CDs regarding their ability to engulf big molecules such as proteins, they might be suitable for formulation and delivery to the retina in the form of small neuroprotective peptides, such as neuropeptide Y (Figure 7). 

## 5. Conclusions

Diabetes is an ever-increasing risk in the Western world, and therefore, diabetes-related microvascular changes are increasingly coming into the focus of scientists and clinicians. In particular, microvascular changes in the retina that can be monitored non-invasively are considered to be warning signs of increased risk of subclinical and clinical stroke, coronary heart disease, heart failure, and nephropathy. Interestingly, microvascular alterations in the retina, heart, and kidney share a similar pathophysiological background, allowing for the search for common biomarkers such as specific pro-inflammatory factors, serum levels of neurotrophins, or a diabetes-associated microRNA signature such as miR-155 or miR-21. With regard to new therapeutic strategies, the treatment of diabetes-associated inflammation, e.g., by specific miRNA therapeutics or targeted delivery of neurotrophic factors into the retina will hopefully provide future effective tools to fight diabetes-related complications.

## Figures and Tables

**Figure 1 biomolecules-12-01113-f001:**
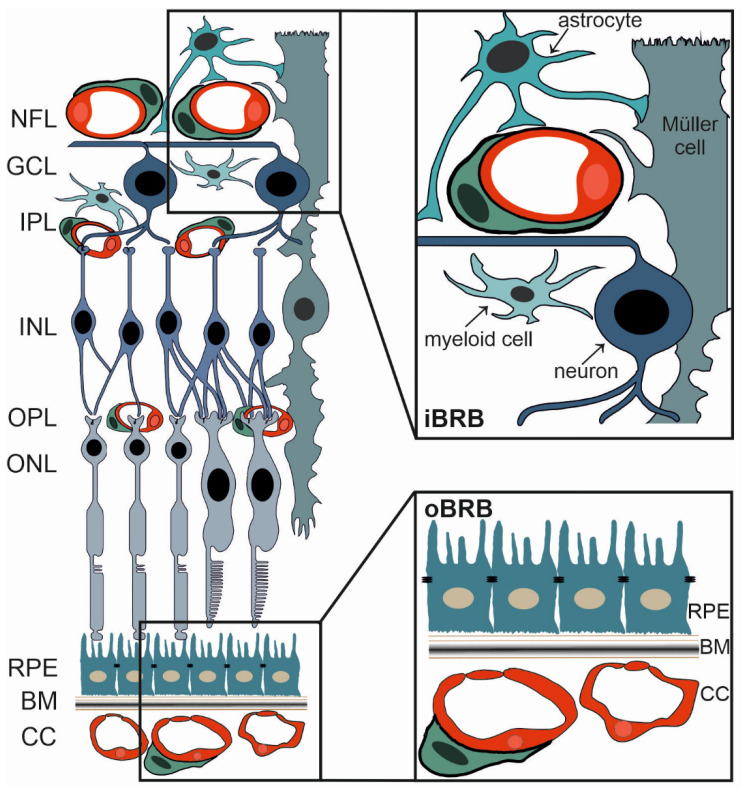
The neurovascular unit and the two blood retinal barriers (BRB). Schematic depicting the morphological architecture of the retina, the Bruch’s membrane (BM) and the choriocapillaris (CC). The inner BRB (iBRB, right upper panel) is formed by tight junctions of endothelial cells (shown in red) of the intraretinal vessels. The neurovascular unit constituting of cells such as pericytes (green), astrocytes, myeloid cells (microglia/macrophages), Müller cells and retinal neurons critically contribute to the integrity of the BRB, too. The outer BRB (oBRB) is formed by the cells of the retinal pigment epithelium (RPE), which are connected by tight junctions. NFL = nerve fiber layer, GCL = ganglion cell layer, IPL = inner plexiform layer, INL = inner nuclear layer, OPL = outer plexiform layer, ONL = outer nuclear layer, RPE = retinal pigment epithelium, BM = Bruch’s membrane, CC = choriocapillaris.

**Figure 2 biomolecules-12-01113-f002:**
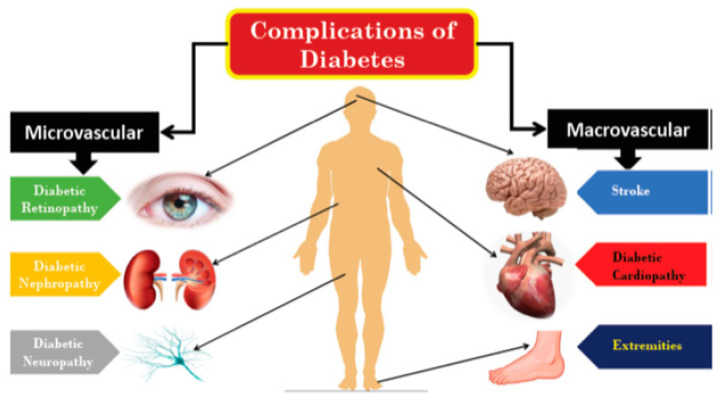
Diabetic microvascular complications have similar etiologies, chronic hyperglycemia initiates diabetic vascular complications through a range of metabolic and structural derangements rooted in the prevalent inflammation, generation of RAGE, ROS production, activation of apoptosis lead to microvascular disease.

**Figure 3 biomolecules-12-01113-f003:**
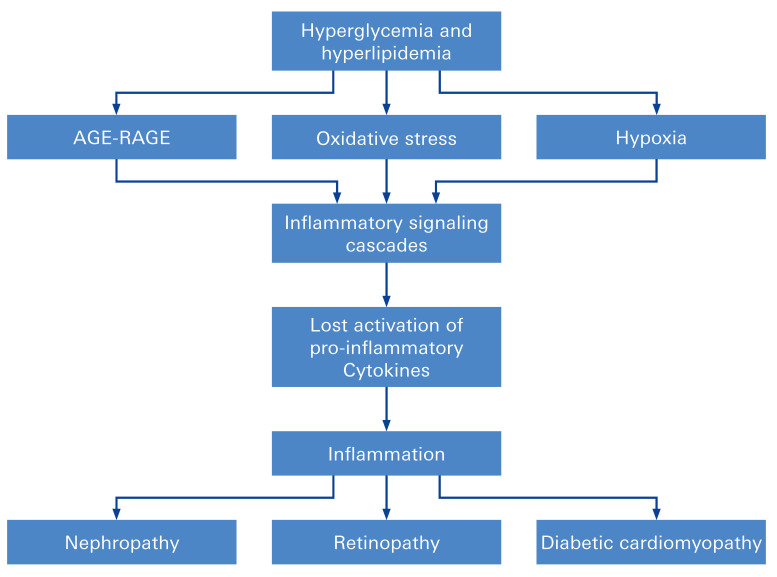
Potential mechanisms for diabetes-associated vascular abnormalities (DAVA). Common pathogenic mechanism as the production of advanced glycation end products (AGE), abnormal activation of signaling cascades (such as protein kinase C [PKC]), elevated production of reactive oxygen species (ROS, oxygen-containing molecules that can interact with other biomolecules and result in damage) [61], and abnormal stimulation of hemodynamic regulation systems (such as the renin-angiotensin system [RAS]).

**Figure 4 biomolecules-12-01113-f004:**
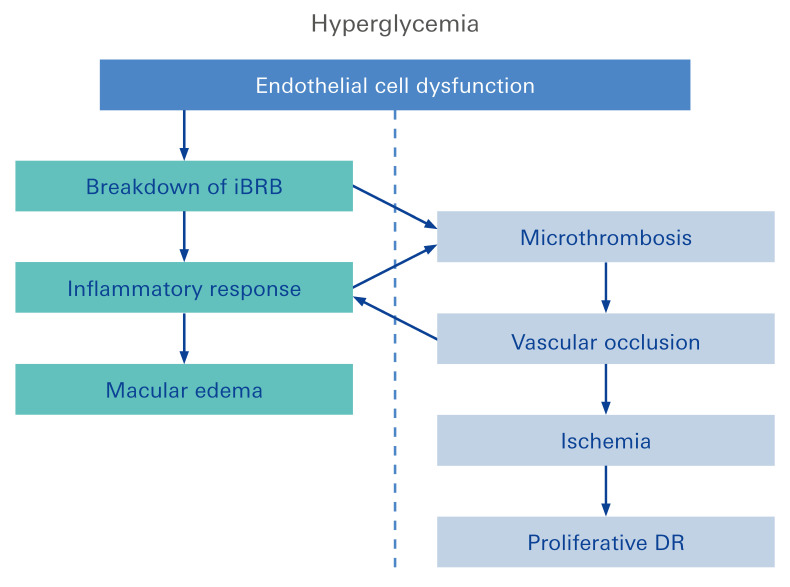
Ocular complications of diabetes: Diabetic retinopathy is a diabetes mellitus-associated retinal condition. The increasing damage to small blood vessels (microangiopathy) causes damage to the retina that is initially unnoticed. However, two complications threaten vision: diabetic macular edema and proliferative diabetic retinopathy. Macular edema is characterized through accumulation of fluid in the macula and seriously distorts vision. The occurrence of retinal/vitreal neovascularization is characteristic of proliferative diabetic retinopathy, the late form of DR, which will lead to blindness if left untreated.

**Figure 5 biomolecules-12-01113-f005:**
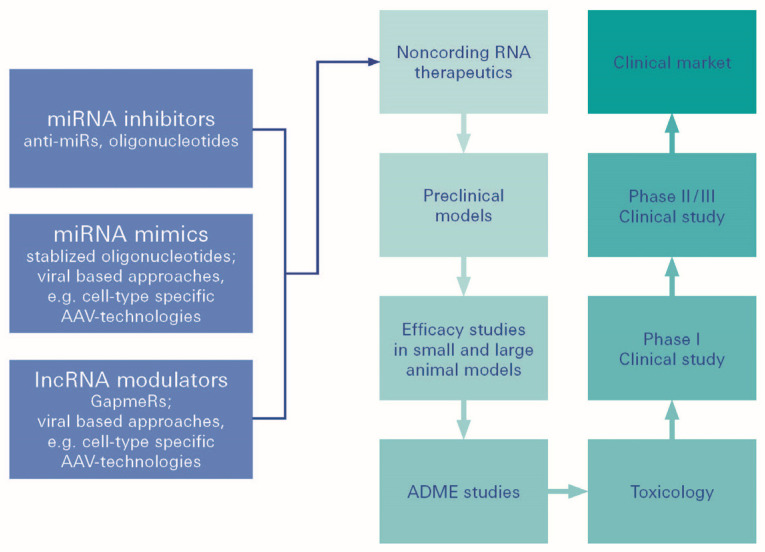
MiRNA therapeutics. MiRNA (miRNA)-based therapeutics can be divided into miRNA mimics and miRNA inhibitors, the so-called antimiRs. miRNA mimics are synthetic double-stranded small RNA molecules that match the corresponding miRNA sequence, and therefore, aim to functionally replenish lacking, dysfunctional, or lost miRNA expression in diseases.

**Figure 6 biomolecules-12-01113-f006:**
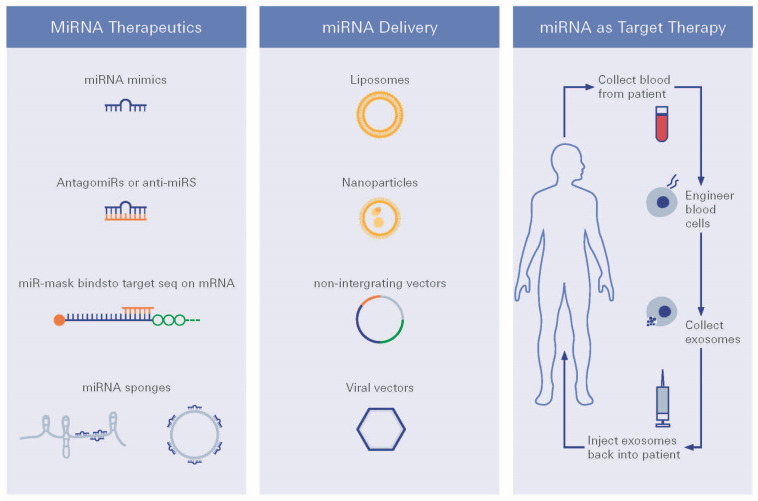
miRNA therapeutic strategies. Different strategies for miRNA targeted therapy, i.e., choice of delivery vehicles and route of delivery are being presented.

**Figure 7 biomolecules-12-01113-f007:**
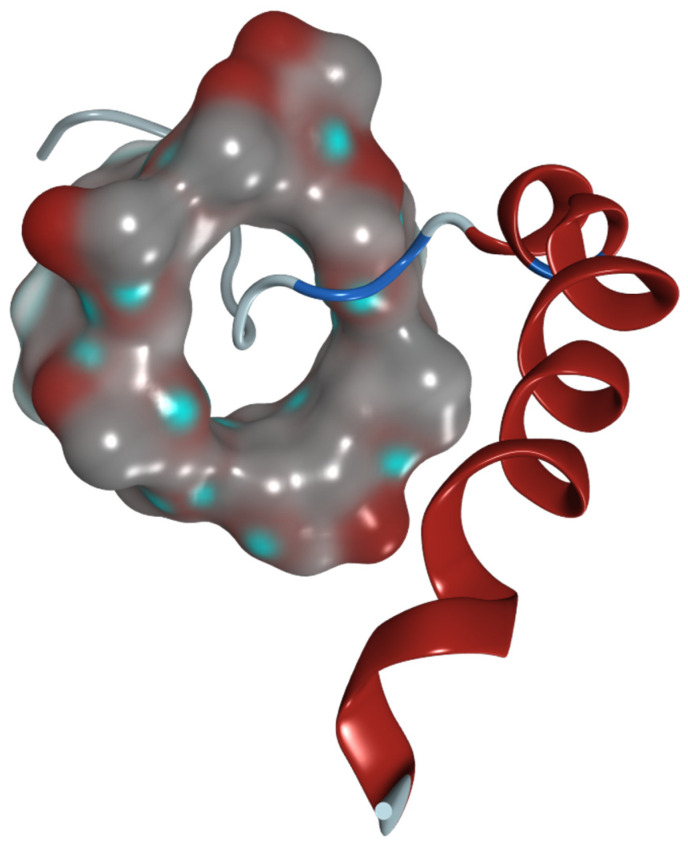
Hypothesized 3D model of methyl-β-cyclodextrin with neuropeptide Y. The HP-β-CD molecule is shown as a molecular surface. The peptide is presented as a ribbon model.

## Data Availability

Not applicable.

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
