# Peer review of "The Protective Effects of Neurotrophins and MicroRNA in Diabetic Retinopathy, Nephropathy and Heart Failure via Regulating Endothelial Function"

_biomolecules, 2022, doi:10.3390/biom12081113_

Round 1

Reviewer 1 Report

Comments and Suggestions for Authors

This review by Shityakov et al. is an interesting review of potential strategies aimed at targeting microvascular complications in diabetes discussing current anti-angiogenic focused therapies and potential future alternative and/ or complementary approaches pointing out the relevance of anti-inflammatory treatments. Beyond that the authors emphasize the importance of monitoring alterations of the retinal microvascular bed as an important predictor of changes in other organs with well-developed special angio-architecture frequently affected by diabetes. Furthermore, the review proposes novel miRNA signatures and neurotrophins as possible surrogates of diabetic microvasculopathy and their potential therapeutic value in fighting ischemic microvascular complications of the retina. The authors address challenges of translation to the clinical application enumerating novel drug-delivery methods. Overall, the review was well written but there are a few concerns that should be addressed.

Major comments for improvement:

1) The Title of the manuscript and the Abstract of the deposited PDF-file appear not to be identical with those of the electronic site provided as a review interface. This means that the uploaded PDF version of the manuscript is an older or newer manuscript version. Alternatively, they belong to two distinct manuscripts. Please, solve this discrepancy.

2) Please mention neurotrophins in the abstract.

3) Figure 2: The anatomical position of aortic bifurcation and the location of kidney in this scheme seem to be quite disproportionate. Please modify it.

4) Figure 3, 4 and 5: Please increase font size within the flowchart for better visibility.

5) Line 362: In order to provide a link to the description of particular miRNAs belonging to the mentioned “194 distinctively expressed miRNAs” in Line 359, please re-edit sentence “MiR-155 therefore plays…” as follows «Among the differentially regulated entities MiR-155 plays ….»

6) “Establishing greater evidence and understaind has moreover even demon- strated that…” The sentence is confusing please re-edit.

7) “4.2.2. Targeted NT delivery to retina”. Please rename this section given that the content of the section discusses not only (neurotrophins) NTs in strict sense but also growth-/neurotrophic factors other than members of the neurotrophin-family. Suggested change «4.2.2. Targeted delivery of neurotrophic factors to the retina».

8) The Manuscript runs out abrupt without spending a few concluding sentences. Please include a brief conclusion at the end of the manuscript text.

Minor comments for improvement:

1. Line 2: “Neurotrohins” spells correctly as Neurotrophins. Please correct it.

2. Line 43: “(Zitat?)”. Typo or a proper reference is missing. Please re-edit sentence.

3. Line 123: “relationsship” spells correctly as relationship. Please correct it.

4. Line 144: “sFigure 3.” Please correct it to Figure 3.

5. Line 152: “reduction of pericytes” is confusing please re-edit to «reduction of pericyte number» or «reduction of pericyte coverage»

6. Line 172: “diabetic associated” should be spelled as diabetes-associated.

7. Line 196: “concomittant” should be spelled as concomitant.

8. Line 218: “HFpEF”. Please define also in the text for better readability.

9. Line 231: “DCM”. Please define also in the text for better readability.

10. Line 262: “…with microalbuminuria (Deckert T, Yokoyama H, Mathiesen E, et al. [84] and 10-fold….” Please re-edit referencing within the sentence.

11. Line 281: “…the signatures of microRNA signatures(miRNA)…”. Please re-edit sentence.

12. Line 330: “therapeuties” should be changed to therapeutics.

13. Line 370: “Remarkaaus morbly”. Please define or check spelling.

14. Line 380: “miRNAtherapeutics” should be changed to miRNA therapeutics.

15. Line 387: “understaind”. Please check spelling.

Author Response

Comments and Suggestions for Authors

This review by Shityakov et al. is an interesting review of potential strategies aimed at targeting microvascular complications in diabetes discussing current anti-angiogenic focused therapies and potential future alternative and/ or complementary approaches pointing out the relevance of anti-inflammatory treatments. Beyond that the authors emphasize the importance of monitoring alterations of the retinal microvascular bed as an important predictor of changes in other organs with well-developed special angio-architecture frequently affected by diabetes. Furthermore, the review proposes novel miRNA signatures and neurotrophins as possible surrogates of diabetic microvasculopathy and their potential therapeutic value in fighting ischemic microvascular complications of the retina. The authors address challenges of translation to the clinical application enumerating novel drug-delivery methods. Overall, the review was well written but there are a few concerns that should be addressed.

Major comments for improvement:

  1. The Title of the manuscript and the Abstract of the deposited PDF-file appear not to be identical with those of the electronic site provided as a review interface. This means that the uploaded PDF version of the manuscript is an older or newer manuscript version. Alternatively, they belong to two distinct manuscripts. Please, solve this discrepancy.A: many thanks for officially pointing this out! Title and abstract had to be changed in frame of the pre-review screen. Please forward your observation to the managing editor in frame of a comment to the editor, so that this discrepancy can be resolved. Many thanks for your support.
  2.  
  3. Please mention neurotrophins in the abstract.
  1. thanks for pointing out, this is mandatory of course. We wrote: “Finally, the use of miRNA, targeted neurotrophin delivery and nanoparticles as novel therapeutic strategies is highlighted., Ll. 28-30

3) Figure 2: The anatomical position of aortic bifurcation and the location of kidney in this scheme seem to be quite disproportionate. Please modify it.

A: many thanks for your comment, this will improve the professionality of the work. We include a redrawn figure 2 according to your advise.

4) Figure 3, 4 and 5: Please increase font size within the flowchart for better visibility.

A: many thanks for your comment, this will improve the quality of the figures. We include a remake of figures 3,4,5 according to your advise.

5) Line 362: In order to provide a link to the description of particular miRNAs belonging to the mentioned “194 distinctively expressed miRNAs” in Line 359, please re-edit sentence “MiR-155 therefore plays…” as follows «Among the differentially regulated entities MiR-155 plays ….»

  1. thanks was replaced “among the differentially regulated entities MiR-155 plays an important role in the inflammation in the heart in diabetes-associated heart failure;”

6) “Establishing greater evidence and understaind has moreover even demon- strated that…” The sentence is confusing please re-edit.

  1. thanks, has been re-edited “understanding”.

As potential blood-borne candidates for DR, the microRNA (miRNA) signatures specifically altered in HFpEF and HFrEF in DM patient blood [88]. Additionally, characteristic neurotrophin levels in the blood experienced pathological changes in the development of DR during routine examination of the eye in diabetics are being proposed [109-113].

And: Moverover, there is a compelling evidence that noncoding RNAs (ncRNAs) may also be expressed in various ways.

7) “4.2.2. Targeted NT delivery to retina”. Please rename this section given that the content of the section discusses not only (neurotrophins) NTs in strict sense but also growth-/neurotrophic factors other than members of the neurotrophin-family. Suggested change «4.2.2. Targeted delivery of neurotrophic factors to the retina».

A: Thanks, headline was changed to “Targeted delivery of neurotrophic factors to the retina”

8) The Manuscript runs out abrupt without spending a few concluding sentences. Please include a brief conclusion at the end of the manuscript text.

  1. Thanks, a conclusion was added and implications were made as requested:
  2.  

“Conclusion:

Diabetes is an ever-increasing risk in the Western world, and therefore diabetes-related microvascular changes are increasingly coming into the focus of scientists and clinicians. In particular, microvascular changes in the retina that can be monitored non-invasively, are considered to be warning signs of increased risk of e.g. subclinical and clinical stroke, coronary heart disease, heart failure, and nephropathy. Interestingly, microvascular alterations in the retina, heart, and kidney share a similar pathophysiological background, allowing the search for common biomarkers such as specific pro-inflammatory factors, serum levels of neurotrophins, or a diabetes-associated microRNA signature such as miR-155 or miR-21. With regard to new therapeutic strategies, the treatment of diabetes-associated inflammation, e.g., by specific miRNA therapeutics or targeted delivery of neurotrophic factors into the retina will hopefully provide future effective tools to fight diabetes-related complications.”

We further recommend edits to language and formatting…

A: The text was meticulously edited and proofread to increase its consistency and flow by a native US American colleague.

Minor comments for improvement:

  1. Line 2: “Neurotrohins” spells correctly as Neurotrophins. Please correct it.

A: thanks, corrected

  1. Line 43: “(Zitat?)”. Typo or a proper reference is missing. Please re-edit sentence. 
  2. A: thanks, corrected
  1. Line 123: “relationsship” spells correctly as relationship. Please correct it. A: thanks, corrected

  1. Line 144: “sFigure 3.” Please correct it to Figure 3. A: thanks, corrected

  1. Line 152: “reduction of pericytes” is confusing please re-edit to «reduction of pericyte number» or «reduction of pericyte coverage» A: thanks, corrected

A: thanks, corrected “the reduction of pericyte numbers“

  1. Line 172: “diabetic associated” should be spelled as diabetes-associated. A: thanks, corrected

  1. Line 196: “concomittant” should be spelled as concomitant. A: thanks, corrected

  1. Line 218: “HFpEF”. Please define also in the text for better readability. A: thanks, HFpEF was defined as heart failure preserved ejection fraction.

  1. Line 231: “DCM”. Please define also in the text for better readability.

A: thanks, DCM was defined as diabetic cardiomyopathy.

  1. Line 262: “…with microalbuminuria (Deckert T, Yokoyama H, Mathiesen E, et al. [84] and 10-fold….” Please re-edit referencing within the sentence.

A: thanks, references were re-edited

  1. Line 281: “…the signatures of microRNA signatures(miRNA)…”. Please re-edit sentence.
  2. thanks was edited
  3. Line 330: “therapeuties” should be changed to therapeutics.

A: thanks, corrected

  1. Line 370: “Remarkaaus morbly”. Please define or check spelling.

A: thanks, corrected “Remarkably,”

  1. Line 380: “miRNAtherapeutics” should be changed to miRNA therapeutics. A: thanks, corrected

  1. Line 387: “understaind”. Please check spelling.

A: thanks, corrected

Author Response

Rebuttal Reviewer 2

Shityakov et al. review the common immunometabolic pathways in diabetic retinopathy (DR),

diabetes-related heart failure, and nephropathy, and discuss potential applications of microRNA

signatures and neurotrophins in diagnostics and therapeutics. This aligns with the aims of the

journal, in particular molecular mechanisms with medical implications. The review is comprehensive

but warrants more refinement in structure as well as in discussion.

With regard to content and structure:

  1. The Introduction would typically begin with the background and relevant definitions, then

transition to the objective(s) of the review at the end of the Introduction. (Consider moving

lines 31 to 39 to the end of the Introduction instead.)

A: Many thanks, the section has been moved as suggested.

  1. There is a lack of clarity in the use of the term of “microvascular”.
  2. In the medical literature, microvascular complications of diabetes are generally: retinopathy, nephropathy, and neuropathy. Cardiac involvement (e.g., heart failure) is considered a macrovascular complication of diabetes. The authors should address/clarify this in the Introduction if possible. Similarly, in Figure 2, the box

Comment: Diabetic cardiomyopathy (DCM) is defined as ventricular dysfunction in the absence of hypertension, coronary artery disease, and valvular heart disease. In 1972, DCM was first reported from the findings of postmortem autopsy of four diabetic patients who manufested heart failure symptoms without coronary artery disease or valvular heart disease (Rubler et al., 1972). The American College of Cardiology Foundation, the American Heart Association (Yancy et al., 2013), and the European Society of Cardiology in collaboration with the European Association for the Study of Diabetes (Ryden et al., 2013) defined DCM as a clinical condition of ventricular dysfunction that occurs in the absence of coronary artery atherosclerosis and hypertension in patients with DM. Thus, DCM is not defined as a macroangiopathy.

Rubler S, Dlugash J, Yuceoglu YZ, Kumral T, Branwood AW, Grishman A. New type of cardiomyopathy associated with diabetic glomerulosclerosis. Am J Cardiol. 1972;30:595–602.

Yancy CW, Jessup M, Bozkurt B, et al; American College of Cardiology Foundation; American Heart Association Task Force on Practice Guidelines. 2013 ACCF/AHA guideline for the management of heart failure: a report of the American College of Cardiology Foundation/American Heart Association Task Force on Practice Guidelines. J Am Coll Cardiol. 2013;62:e147–e239.

Ryden L, Grant PJ, Anker SD, Berne C, Cosentino F, Danchin N, Deaton C, Escaned J, Hammes HP, Huikuri H, Marre M, Marx N, Mellbin L, Ostergren J, Patrono C, Seferovic P, Uva MS, Taskinen MR, Tendera M, Tuomilehto J, Valensi P, Zamorano JL, Zamorano JL, Achenbach S, Baumgartner H, Bax JJ, Bueno H, Dean V, Deaton C, Erol C, Fagard R, Ferrari R, Hasdai D, Hoes AW, Kirchhof P, Knuuti J, Kolh P, Lancellotti P, Linhart A, Nihoyannopoulos P, Piepoli MF, Ponikowski P, Sirnes PA, Tamargo JL, Tendera M, Torbicki A, Wijns W, Windecker S, De Backer G, Sirnes PA, Ezquerra EA, Avogaro A, Badimon L, Baranova E, Baumgartner H, Betteridge J, Ceriello A, Fagard R, Funck-Brentano C, Gulba DC, Hasdai D, Hoes AW, Kjekshus JK, Knuuti J, Kolh P, Lev E, Mueller C, Neyses L, Nilsson PM, Perk J, Ponikowski P, Reiner Z, Sattar N, Schachinger V, Scheen A, Schirmer H, Stromberg A, Sudzhaeva S, Tamargo JL, Viigimaa M, Vlachopoulos C, Xuereb RG; Authors/Task Force Members; ESC Committee for Practice Guidelines (CPG); Document Reviewers. ESC Guidelines on diabetes, pre-diabetes, and cardiovascular diseases developed in collaboration with the EASD: the Task Force on diabetes, pre-diabetes, and cardiovascular diseases of the European Society of Cardiology (ESC) and developed in collaboration with the European Association for the Study of Diabetes (EASD). Eur Heart J. 2013;34:3035–3087.

In the introduction, we added: Diabetic cardiomyopathy (DCM) is defined as ventricular dysfunction in the absence of hypertension, coronary artery disease, and valvular heart disease. In 1972, DCM was first reported from the findings of postmortem autopsy of four diabetic patients who manufested heart failure symptoms without coronary artery disease or valvular heart disease (Rubler et al., 1972). The American College of Cardiology Foundation, the American Heart Association (Yancy et al., 2013), and the European Society of Cardiology in collaboration with the European Association for the Study of Diabetes (Ryden et al., 2013) defined DCM as a clinical condition of ventricular dysfunction that occurs in the absence of coronary artery atherosclerosis and hypertension in patients with DM.

In Fig.2, we changed „Diabetic microvascular and macrovascular complications“ to „Diabetic microvascular complications“

“Diabetic cardiomyopathy” is listed under the heading “Microvascular complications”. This can be misleading.

Comment: The pathological characteristic of diabetes-related vascular complications is damage to the microcirculation throughout the body. Unique examples of microvascular complications inlcude diabetic nephropathy and retinopathy (Pappachan et al., 2013). Impaired coronary microvasculature is frequently observed in patients with T2DM, insulin resistance, and DCM (Factor et al., 1984; Adameova and Dhalla, 2014). This defect is caused by reduced levels of bioavailable nitric oxide (Zhou et al., 2010). In coronary vascular smooth muscle cells, nitric oxide activates both kinases and guanylyl cyclase, which is required for coronary relaxation (Hayden et al., 2012). Under conditions of reduced insulin sensitivity, both increased nitric oxide degradation and reduced nitric oxide production occur. Reduced capillary length density and hyaline-related changes in the medial arteriolar layers are observed in the cardiac circulation of diabetes mellitus patients (Fang et al., 2004; Campbell et al., 2011). The reduced blood supply resulting from microcirculatory dysfunction affecting the vasa vasorum in diabetes mellitus further damages the medium and small arterioles of the diabetic heart. Perivascular fibrosis and interstitial changes, the formation of microaneurysms in small vessels, and thickening of the capillary basement membrane are other vascular disorders that cause cardiac microvascular ischemia in diabetes mellitus. Ischemia contributes to myocardial fibrosis, stiffness, and dysfunction in DCM (Lee and Kim, 2017). It is correct to assume that DCM is due to microangiopathy.

Pappachan JM, Varughese GI, Sriraman R, Arunagirinathan G. Diabetic cardiomyopathy: pathophysiology, diagnostic evaluation and management. World J Diabetes 2013;4:177-189.

Factor SM, Minase T, Cho S, Fein F, Capasso JM, Sonnenblick EH. Coronary microvascular abnormalities in the hypertensive-diabetic rat: a primary cause of cardiomyopathy? Am J Pathol 1984;116:9-20.

Adameova A, Dhalla NS. Role of microangiopathy in diabetic cardiomyopathy. Heart Fail Rev 2014;19:25-33.

Zhou X, Ma L, Habibi J, et al. Nebivolol improves diastolic dysfunction and myocardial remodeling through reductions in oxidative stress in the Zucker obese rat. Hypertension 2010;55:880-888.

Hayden MR, Habibi J, Joginpally T, Karuparthi PR, Sowers JR. Ultrastructure study of transgenic Ren2 rat aorta. Part 1: rndothelium and intima. Cardiorenal Med 2012;2:66-82.

Fang ZY, Prins JB, Marwick TH. Diabetic cardiomyopathy: evidence, mechanisms, and therapeutic implications. Endocr Rev 2004;25:543-567.

Campbell DJ, Somaratne JB, Jenkins AJ, et al. Impact of type 2 diabetes and the metabolic syndrome on myocardial structure and microvasculature of men with coronary artery disease. Cardiovasc Diabetol 2011;10:80.

Lee WS, Kim J. Diabetic cardiomyopathy: where we are and where we are going. Korean J Intern Med. 2017;32:404-421.

  1. In line 75, Section 2 heading: “towards a microvascular signature for …” What is meant by a “microvascular signature”? A signature found only in the microvasculature?

Comment: Thank you. This is very important issue. The heading is not clear. We changed: „Eyes, cardiac and renal microvasculature – towards a microvascular signature for end organ damage in diabetic retinopathy, diabetic nephropathy and heart failure“ to „Eyes, renal and cardiac microvasculature – From a microvascular signature to end organ damage in diabetic retinopathy, diabetic nephropathy and diabetic cardiomyopathy“.

  1. There is a lack of clarity in the link that the authors are attempting to establish between the three conditions.
  2. In lines 31 to 34, “… the elucidation of common immunometabolic pathways to endorgan damage in diabetes-related heart failure, diabetic retinopathy (DR) and nephropathy, respectively, could greatly help to facilitate early diagnosis, progression analysis, and therapy development.” – Early diagnosis, progression analysis, and therapy development of what? All three conditions? (Or changes in DR being a marker for the other two conditions?)

Comment: The reviewer is right. We changed: „, and therapy development“ to „, and therapy development of all three condiitons“.

  1. In lines 34 to 37, “Being a uniquely specific and non-invasive assessable measure of diabetic microvascular damage, analyses of DR–related early microvascular changes in the eye could help to detect microvascular disease in otherwise asymptomatic patients.” – This is confusing because it is not established what is meant by“microvascular disease”. The term can be assumed to mean diabetic retinopathy, nephropathy, and neuropathy (see point 2a above).

Comment: The pathological characteristic of diabetes-related vascular complications is damage to the microcirculation throughout the body. Unique examples of microvascular complications inlcude diabetic nephropathy and retinopathy (Pappachan et al., 2013). Impaired coronary microvasculature is frequently observed in patients with T2DM, insulin resistance, and DCM (Factor et al., 1984; Adameova and Dhalla, 2014). This defect is caused by reduced levels of bioavailable nitric oxide (Zhou et al., 2010).

In the introduction, we added: „Impaired coronary microvasculature is frequently observed in patients with T2DM, insulin resistance, and DCM (Factor et al., 1984; Adameova and Dhalla, 2014).“

Factor SM, Minase T, Cho S, Fein F, Capasso JM, Sonnenblick EH. Coronary microvascular abnormalities in the hypertensive-diabetic rat: a primary cause of cardiomyopathy? Am J Pathol 1984;116:9-20.

Adameova A, Dhalla NS. Role of microangiopathy in diabetic cardiomyopathy. Heart Fail Rev 2014;19:25-33.

In the introduction, we changed: „DR–related early microvascular changes in the eye could help to detect microvascular disease in otherwise asymptomatic patients.“ to „DR–related early microvascular changes in the eye could help to detect microvascular associated diseases such as such as diabetic nephropathy and DCM in otherwise asymptomatic patients.“

  1. In lines 38 to 39, “… miRNA signatures (miRNA) and characteristic neurotrophin levels in patient blood might have the potential serving as blood-borne biomarker candidates.” – Blood-borne biomarker candidates for what?

Comment: Thank you. We changed: „blood-borne biomarker candidates“ to „blood-borne biomarker candidates for DR“.

  1. The authors seem to suggest that DR (various parts of the article) and nephropathy (Section 2.4) are risk factors for cardiomyopathy. However, they also highlight that “the lack of early biomarkers coupled with the fact that the earlier stages of diabetic cardiomyopathy and nephropathy are mostly asymptomatic makes detection a clinical challenge” (lines 270 to 272), and “… as the retinal vessels can easily be visualized through funduscopy and/or angiography [31,72,73], microvascular alterations caused by systemic diseases such as diabetes can easily be observed in retinal blood vessels and allow predictions on the vascular status of other organs, e.g., the heart, the kidney and/or the brain.” (lines 210 to 213) Do the authors envision microvascular changes in DR being a marker for both diabetic nephropathy and diabetic cardiomyopathy? If so, please make it clear from the outset / Introduction.

Comment: Thnak you. In the introduction, we changed: „DR–related early microvascular changes in the eye could help to detect microvascular disease in otherwise asymptomatic patients.“ to „DR–related early microvascular changes in the eye could help to detect microvascular associated diseases such as such as diabetic nephropathy and DCM in otherwise asymptomatic patients.“

  1. Nephropathy is not adequately discussed.
  2. In the Introduction, no reference is made to nephropathy, despite it being one of the three conditions that are the focus of this review.

Comment: The reviewer is right. Diabetes-related microvascular complications such as diabetic nephropathy constitute a significant public health problem. Diabetic nephropathy is a leading cause of chronic kidney disease and end-stage kidney disease (Alicic et al., 2017). Furthermore, most of the excess mortality risk observed in patients with diabetes may be related to the presence of diabetic nephropathy (Afkarian et al., 2013). SGLT2i have recently emerged as a new class of oral glucose-lowering agents with pleiotropic effects including reduction in cardiovascular and kidney outcomes among patients with T2D (Perkovic et al., 2019). In a recent meta-anlysis, SGLT2i slowed estimated glomerular filtration rate decline, lowered albuminuria progression, improved adverse renal endpoints (Wang et al., 2019). Nephropathy is important microvascular compications of T2DM.

Alicic RZ, Rooney MT, Tuttle KR. Diabetic kidney disease: challenges, progress, and possibilities. Clin J Am Soc Nephrol. 2017;12:2032–45.

Afkarian M, Sachs MC, Kestenbaum B, Hirsch IB, Tuttle KR, Himmelfarb J, et al. Kidney disease and increased mortality risk in type 2 diabetes. J Am Soc Nephrol. 2013;24:302–8.

Perkovic V, Jardine MJ, Neal B, Bompoint S, Heerspink HJL, Charytan DM, et al. Canaglifozin and renal outcomes in type 2 diabetes and nephropathy. N Engl J Med. 2019;380:2295–306.

Wang C, Zhou Y, Kong Z, Wang X, Lv W, Geng Z, Wang Y. The renoprotective effects of sodium-glucose cotransporter 2 inhibitors versus placebo in patients with type 2 diabetes with or without prevalent kidney disease: A systematic review and meta-analysis. Diabetes Obes Metab. 2019;21:1018-1026.

In the introduction, we added: „Diabetes-related microvascular complications such as diabetic nephropathy constitute a significant public health problem. Diabetic nephropathy is a leading cause of chronic kidney disease and end-stage kidney disease (Alicic et al., 2017). Furthermore, most of the excess mortality risk observed in patients with diabetes may be related to the presence of diabetic nephropathy (Afkarian et al., 2013). SGLT2i have recently emerged as a new class of oral glucose-lowering agents with pleiotropic effects including reduction in cardiovascular and kidney outcomes among patients with T2D (Perkovic et al., 2019). In a recent meta-anlysis, SGLT2i slowed estimated glomerular filtration rate decline, lowered albuminuria progression, improved adverse renal endpoints (Wang et al., 2019). Nephropathy is important microvascular compications of T2DM.“

  1. Furthermore, the only instance where nephropathy is discussed is in one subsection, “2.4 Diabetic nephropathy”.

Comment: The reviewer is right. The issue between DR and nephropathy was little mentioned. The progression of DR and diabetic nephropathy for patients with T2DM was discordant. The Renal Insufficiency and Cardiovascular Events study found that the progression of diabetic nephropathy did not affect 41.4% of T2DM patients with advanced DR (Pennno et al., 2012). Glycemic variability over the long term was found not to affect the progression of DR but could predict the presence of DN (Pennno et al., 2013). These results could be explained by the pathogenesis of DR and diabetic nephropathy which could be affected by different risk factors. Moreover, the previous meta-analysis has already found that DR should be regarded as a useful status for diagnosing and predicting diabetic nephropathy for patients with T2DM (He et al., 2013). Another recent meta-analysis found significant associations between DR and subsequent DN risk for patients with T2DM (Li et al., 2021).Thus, the significant relatinsip between DR and nephropathy have been shown until now.

Penno G, Solini A, Zoppini G, et al. Rate and determinants of association between advanced retinopathy and chronic kidney disease in patients with type 2 diabetes: the Renal Insufficiency and Cardiovascular Events (RIACE) Italian multicenter study. Diabetes Care. 2012;35:2317-2323.

Penno G, Solini A, Bonora E, et al. HbA1c variability as an independent correlate of nephropathy, but not retinopathy, in patients with type 2 diabetes: the renal insufficiency and cardiovascular events (RIACE) Italian multicenter study. Diabetes Care. 2013;36:2301-2310.

He F, Xia X, Wu XF, et al. Diabetic retinopathy in predicting diabetic nephropathy in patients with type 2 diabetes and renal disease: a meta-analysis. Diabetologia. 2013;56:457-466.

Li Y, Su X, Ye Q, Guo X, Xu B, Guan T, Chen A. The predictive value of diabetic retinopathy on subsequent diabetic nephropathy in patients with type 2 diabetes: a systematic review and meta-analysis of prospective studies. Ren Fail. 2021;43:231-240.

In the“2.4 Diabetic nephropathy”, we added: „The progression of DR and diabetic nephropathy for patients with T2DM was discordant. The Renal Insufficiency and Cardiovascular Events study found that the progression of diabetic nephropathy did not affect 41.4% of T2DM patients with advanced DR (Pennno et al., 2012). Glycemic variability over the long term was found not to affect the progression of DR but could predict the presence of DN (Pennno et al., 2013). These results could be explained by the pathogenesis of DR and diabetic nephropathy which could be affected by different risk factors. Moreover, the previous meta-analysis has already found that DR should be regarded as a useful status for diagnosing and predicting diabetic nephropathy for patients with T2DM (He et al., 2013). Another recent meta-analysis found significant associations between DR and subsequent DN risk for patients with T2DM (Li et al., 2021).“

  1. Oddly, the heading of subsection 4.2 is “4.2. Identification of a common microRNA signature in damage in diabetic retinopathy, diabetic nephropathy and heart failure”. However, the subsequent paragraphs make no mention of nephropathy, discussing only DR and heart failure.

Comment: Thank you. These are very important issue. We added the relationships of micro RNA with diabetic nephropathy and retinopathy were summarized as follows. Diabetic nephropathy; miR-155 negatively regulates the expression of target gene E26 transformation-specific Sequence 1 (ETS-1) and its downstream factors VCAM-1, MCP-1 and cleaved caspase-3, thus mediating the inflammatory response and apoptosis of human renal glomerular endothelial cells (He et al., 2022). The miR-155 expression was increased in renal tissues of DN patients, and mainly expressed in glomerular vascular endothelial cells, mesangial cells and renal tubule interstitium (Mikhail et al., 2017). The results of luciferase reporter gene showed that ETS-1 may be a potential target gene of miR-155. Further detection of serum miRNAs in diabetic patients showed abnormal expression of miR-155 in diabetic patients compared with healthy controls, and the expression of miR-155 was significantly different in microproteinuria and macroproteinuria groups, and was positively correlated with eGFR in diabetic nephropathy patients and negatively correlated with urinary protein excretion rate (Mikhail et al., 2017). Triptolide up-regulated BDNF by inhibiting miR-155-5p, thus inhibiting oxidative stress and inflammatory damage and alleviating podocyte injury in diabetic nephropathy mice (Gao et al., 2022). The microRNA-195 promotes apoptosis of podocytes under high-glucose conditions via enhanced caspase cascades for BCL2 insufficiency (Chen et al., 2011). The abated microRNA-195 expression protected mesangial cells from apoptosis, suggesting that the antiapoptosis in a microRNA-regulated manner may play an important role in the early stages of diabetic nephropathy (Chen et al., 2012). miR-455-3p suppresses renal fibrosis through repression of ROCK2 expression in diabetic nephropathy (Wu et al., 2018). In the DN mouse model, Circ_0000491 knockdown inhibited high glucose-induced apoptosis, inflammation, oxidative stress, and fibrosis in SV40-MES13 cells by regulating miR-455-3p/Hmgb1 axis (Wang et al., 2022). Diabetic retinopathy; In T2DM retinopathy, miR-155 plays an important role in the pathogenesis of T2DM retinopathy by regulating the Treg cells with TGF-β. rs767649 polymorphism in the pre-MIR155 gene is associated with DR in T2DM and that the miR-155 plasma levels might be associated with T2DM (Polina et al., 2019). miR-155-5p expression was significantly upregulated in human retinal microvascular endothelial cells induced by high glucose. After inhibiting the expression of miR-155-5p, cell proliferation, angiogenesis and VEGF protein levels were significantly downregulated, whereas miR-155-5p mimics had the opposite effect. miR-155-5p is closely associated with diabetic macular edema and is a potential target for refractory diabetic macular edema treatment (He et al., 2021). miR-195 regulates SIRT1-mediated tissue damage in DR (Mortuza et al., 2014). miR-195 knockdown led to the downregulation of the mRNA and protein expression levels of BAX and the upregulation of the mRNA and protein expression levels of SIRT1 and BCL-2 as well as improvement in cell growth and a decrease in the apoptosis rate (Shan et al., 2022). miR-195 is overexpressed in DR, and its targeted relationship with SIRT1 inhibits the growth of cells in the retina and accelerates apoptosis (Shan et al., 2022). miR-455-5p ameliorates high glucose-induced apoptosis, oxidative stress and inflammatory via targeting SOCS3 in retinal pigment epithelial cells (Chen et al., 2019).

He K, Chen Z, Zhao J, He Y, Deng R, Fan X, Wang J, Zhou X. The role of microRNA-155 in glomerular endothelial cell injury induced by high glucose. Mol Biol Rep. 2022;49:2915-2924.

Mikhail M, Vachon PH, D’Orleans-Juste P, et al. Role of endothelin-1 and its receptors ETA and ETB in the survival of human vascular endothelial cells. Can J Physiol Pharmacol. 2017;95:1298-1305.

Gao J, Liang Z, Zhao F, Liu X, Ma N. Triptolide inhibits oxidative stress and inflammation via the microRNA-155-5p/brain-derived neurotrophic factor to reduce podocyte injury in mice with diabetic nephropathy. Bioengineered. 2022;13:12275-12288.

Chen YQ, Wang XX, Yao XM, Zhang DL, Yang XF, Tian SF, Wang NS. MicroRNA-195 promotes apoptosis in mouse podocytes via enhanced caspase activity driven by BCL2 insufficiency. Am J Nephrol. 2011;34:549-559.

Chen YQ, Wang XX, Yao XM, Zhang DL, Yang XF, Tian SF, Wang NS. Abated microRNA-195 expression protected mesangial cells from apoptosis in early diabetic renal injury in mice. J Nephrol. 2012;25:566-576.

Wu J, Liu J, Ding Y, Zhu M, Lu K, Zhou J, Xie X, Xu Y, Shen X, Chen Y, Shao X, Zhu C. MiR-455-3p suppresses renal fibrosis through repression of ROCK2 expression in diabetic nephropathy. Biochem Biophys Res Commun. 2018;503:977-983.

Wang J, Yang S, Li W, Zhao M, Li K. Circ_0000491 Promotes Apoptosis, inflammation, oxidative stress, and fibrosis in high glucose-induced mesangial cells by regulating miR-455-3p/Hmgb1 Axis. Nephron. 2022;146:72-83.

Yang TT, Song SJ, Xue HB, Shi DF, Liu CM, Liu H. Regulatory T cells in the pathogenesis of type 2 diabetes mellitus retinopathy by miR-155. Eur Rev Med Pharmacol Sci. 2015;19:2010-2015.

Polina ER, Oliveira FM, Sbruzzi RC, Crispim D, Canani LH, Santos KG. Gene polymorphism and plasma levels of miR-155 in diabetic retinopathy. Endocr Connect. 2019;8:1591-1599.

He J, Zhang R, Wang S, Xie L, Yu C, Xu T, Li Y, Yan T. Expression of microRNA-155-5p in patients with refractory diabetic macular edema and its regulatory mechanism. Exp Ther Med. 2021;22:975.

Mortuza R, Feng B, Chakrabarti S. miR-195 regulates SIRT1-mediated changes in diabetic retinopathy. Diabetologia. 2014;57:1037-1046.

Shan L, Zhang H, Han Y, Kuang R. Expression and mechanism of microRNA 195 in diabetic retinopathy. Endocr J. 2022;69:529-537.

Chen P, Miao Y, Yan P, Wang XJ, Jiang C, Lei Y. MiR-455-5p ameliorates HG-induced apoptosis, oxidative stress and inflammatory via targeting SOCS3 in retinal pigment epithelial cells. J Cell Physiol. 2019;234:21915-21924.

  • In the manuscript text body, all these references were included by use of EndNote literature managing program.

  1. In the Introduction lines 53 to 65, what is the point of listing all the drugs that did or did not

reduce heart failure risk? Significance is not explained.

  1. Thanks , the section was redacted.

  1. Based on the title of the review, neutrophins and microRNA seem to be key terms but are

not appropriately introduced in the first section (Introduction).

  1. Thanks, this will greatly improve the quality of the review manuscript. We added a paragraph for both topics, microRNA and neurotrophins.

  1. In lines 342 to 344, “Recently, it was shown that miRNAs as a more stable form,

circulate in the blood. This would suggest that circulating miRNAs are used as

biomarkers for cardiovascular diseases such as DR and diabetes-associated heart

failure.” – Second sentence cannot be concluded from first sentence. Also are

miRNAs really used as biomarkers for DR and diabetes-associated heart failure

currently?

  1. Recently, it was shown that miRNAs as a more stable form, circulate for a longer period of time as conventional mRNA in the blood. Therefore, these circulating miRNAs might be detected after mRNA is degraded, giving them an advantage to be used as a biomarkers for cardiovascular diseases such as DR and diabetes-associated heart failure.

  1. Certain conclusions seem to be drawn without a sound basis or reasoning.
  2. In lines 64 to 67, “Interestingly, emerging evidence suggests that DR may share common genetic linkages with systemic vascular complications in DM. As a result, DR might reflect a widespread microcirculatory disease not only in the eye but also vital organs elsewhere in the body [9].” – Evidence of common genetic linkages alone does not directly mean that DR signifies microcirculatory disease in other vital organs?

Comment: The reviewer is correct. We changed: „DR may share common genetic linkages with systemic vascular complications in DM“ to „DR may share common pathophysiology with systemic vascular complications in DM“.

  1. In lines 242 to 243, “As pointed out, recent results suggest that DR and diabetesassociated heart failure have a similar pathophysiological, inflammatory, microvascular background. The occurrence of DR is therefore an independent predictor of heart failure.” – Again, first sentence does not directly imply second sentence. Furthermore, the term “independent predictor” would usually imply that there is a significant association, even after controlling for covariates (i.e., in a multivariable model). This seems to be a finding in only one study (Cheung et al., 2008) which the authors have cited, but not when making this claim. Perhaps this citation should be added when making this statement. The authors may also wish to consider making it clear in the statement itself that it is a finding of this (one) study, rather than presenting it as an established fact. If there are additional studies supporting this claim, please cite them as wel

Comment: Thank you. The second sentence was too strong because the results was not sufficient to support the notion that „the relationship between microvascular disease and HFpEF was stronger than that between macrovascular disease and HFpEF“. We omitted this sentence. On the othe hand, data from the Multiethnic Study of Atherosclerosis (MESA) also showed that microvascular complications were associated with more concentric hypertrophy on echocardiography, a hallmark of HFpEF (Cheung et al., 2007). In the PROMIS-HFpEF study, a high prevalence of coronary microvascular dysfunction in HFpEF was shown in the absence of unrevascularized macrovascular coronary artery disease (Shah, et al., 2018). These should be added as the reviewer indicated.

In the section „2.3. Heart failure in Diabetes mellitus“, we added: „Data from the Multiethnic Study of Atherosclerosis (MESA) also showed that microvascular complications were associated with more concentric hypertrophy on echocardiography, a hallmark of HFpEF (Cheung et al., 2007). In the PROMIS-HFpEF study, a high prevalence of coronary microvascular dysfunction in HFpEF was shown in the absence of unrevascularized macrovascular coronary artery disease (Shah, et al., 2018).“

Cheung N, Bluemke DA, Klein R, et al. Retinal arteriolar narrowing and left ventricular remodeling: the multi-ethnic study of atherosclerosis. J Am Coll Cardiol 2007;50:48-55.

Shah SJ, Lam CSP, Svedlund S, et al. Prevalence and correlates of coronary microvascular dysfunction in heart failure with preserved ejection fraction: PROMIS-HFpEF. Eur Heart J 2018;39:3439-3450.

  1. The paragraph from lines 386 to 391 is confusing.

  1. In lines 387 to 388, “Establishing greater evidence and understaind has moreover

even demonstrated” – please check spelling. Expression is convoluted, please

rephrase. A: thanks for pointing out, was corrected.

  1. In lines 388 to 389, not sure what is meant by “DR expression may play a vital role in

the development of DR.”

  1. In lines 389 to 390, not sure what is meant by “Amongst the non-coding RNAs,

ncRNAs, besides miRNAs and also the long ncRNAs (lncRNAs)”. Why are ncRNAs

described as “amongst” non-coding RNAs, if they are the same entity?

A: we rephrased the sentences as

  1. In Section 3 on current treatment options, only treatment for DR is discussed. Is the focus of

the review on treatment of DR only? This seems disparate from the preceding sections, in

which the authors (supposedly) propose that markers in DR may predict heart failure and

nephropathy.

  1. The reviewer is right. We added a paragraph: About adipose tissue inflammation and heart failure, several clinical studies have shown an association between systemic insulin resistance, inflammation and heart failure. For example, glucose metabolism is impaired in patients without type 2 diabetes who have idiopathic dilated cardiomyopathy [129].

Diabetic nephropathy has so far not been traditionally considered an inflammatory disease. However, recent studies have shown that kidney inflammation is crucial in promoting the development and progression of Diabetic nephropathy [130].

  1. Again, when discussing future / potential diagnostics and therapeutics, the interrelation

between the three conditions is not clearly established:

  1. In Section 4.1 on targeting inflammatory mechanisms, only diabetes and DR are

discussed. Does this mean that this approach is only for DR, and not heart failure

and nephropathy?

  1. In Section 4.2 on microRNA signatures, only DR and heart failure are discussed. Does

this mean that this approach is only for DR and heart failure, and not nephropathy?

  1. thanks for pointing this out. So far, similar evidence for these approaches could not be found in the literature, thus we limited our exptrapolation to DR and heart failure.
  2.  
  1. In Sections 4.2.1 and 4.2.2 on neurotrophins, only DR is discussed. Does this mean

that this approach is only for DR, and not heart failure and nephropathy?

  1. Note: Nephropathy is not discussed at all.
  2. To our knowledge, so far evidence points to supportive effects of neurotrophins in DR, or other diabetes-associated complications as neuropathy, cutaneous innervation.

While some evidence is given in the literature, that neurotrophins might be benficial in cardiac disease, these include atherosclerosis, hypertension, diabetes, acute myocardial infarction, and heart failure, no direct relation to specifically diabetes –associated heart failure could be found. In many of these conditions, altered expression of neurotrophins and/or neurotrophin receptors has direct effects on vascular endothelial and smooth muscle cells in addition to effects on nerves that modulate vascular resistance and cardiac function.

[The biology of neurotrophins: cardiovascular function. Emanueli C, Meloni M, Hasan W, Habecker BA.Emanueli C, et al. Among authors: habecker ba. Handb Exp Pharmacol. 2014;220:309-28. doi: 10.1007/978-3-642-45106-5_12.]

This might progress in the future as the field of neurocardiology will develop more.

  1. In Figure 5 caption (lines 393 to 397), while it is state that miRNA-based therapeutics are

divided into miRNA mimics and miRNA inhibitors, only miRNA mimics are elaborated upon,

but not miRNA inhibitors. Consider adding a line for miRNA inhibitors as well. Additionally, a

third box, “lncRNA modulators” is shown in the figure, but lacks any mention in the caption.

Unclear: is there a reason for inclusion in the figure but omission in the figure caption?

  1. In lines 454 to 457, please cite which studies used samples from diabetic patients in late

stages of disease, and which studies were collected at earlier stages. (Or is this just

speculation?)

  1. The reviewer is right: Indeed, this sentence is speculation/discussion. In contrast, aqueous humor or vitreous samples can be obtained with appropriate ethic applications during routine surgery and therefore, this can well be done at earlier time points of the disease.
  2. We came to that idea, as, at least under German law, it is impossible to remove retinal tissue from a patient during routine surgery. Consequently, we assume that if retinal tissue was examined, it was obtained post-mortem and this, assuming a natural cause of death, will usually be associated with rather more advanced diabetes.

  1. The ending of the article is abrupt. Please add a “Conclusions” section to summarize the key

points discussed in the review as well as future directions.

  1. The implications for research and clinical practice are not clearly highlighted.

  1. Thanks, a conclusion was added and implications were made as requested:
  2.  

“Conclusion:

Diabetes is an ever-increasing risk in the Western world, and therefore diabetes-related microvascular changes are increasingly coming into the focus of scientists and clinicians. In particular, microvascular changes in the retina that can be monitored non-invasively, are considered to be warning signs of increased risk of e.g. subclinical and clinical stroke, coronary heart disease, heart failure, and nephropathy. Interestingly, microvascular alterations in the retina, heart, and kidney share a similar pathophysiological background, allowing the search for common biomarkers such as specific pro-inflammatory factors, serum levels of neurotrophins, or a diabetes-associated microRNA signature such as miR-155 or miR-21. With regard to new therapeutic strategies, the treatment of diabetes-associated inflammation, e.g., by specific miRNA therapeutics or targeted delivery of neurotrophic factors into the retina will hopefully provide future effective tools to fight diabetes-related complications.”

  1. Thanks for these helpful comments, # 12-14 were modified as suggested.

  1. At times, the review seems to be stitching together findings from different studies without

further synthesis and/or substantiation of a central point. To enhance clarity, the manuscript

should clearly answer the following questions: What has already been found? What are the

next steps / future directions to reach potential new findings or clinical applications?

  1. Is Figure 1 (or any of the other figures) reproduced or adapted from a previously published

image? If so, please include a copyright attribution in the figure caption, indicating the origin

of the reproduced or adapted material, in addition to a reference list entry for the work.

We further recommend edits to language and formatting, which is important for greater clarity of

expression and to ensure proper conveyance of the authors’ intended meaning:

A: thanks we had another native English check by our collegue Mr. Todd Johnsen and hope to have removed all unclear parts

  1. Please check for correct spelling throughout the article.
  2. Please check the spelling in the title – it should be “Neurotrophins” instead of

“Neurotrohins”. A. Thanks, corrected

  1. In line 123, check spelling: replace “relationsship” with “relationship”. A. Thanks, corrected

  1. In line 196, check spelling: replace “concomittant" with “concomitant”. A. Thanks, corrected

  1. In line 327, check spelling: replace “vitreous” with “vitreous”. A. Thanks, corrected

  1. In line 356, check spelling: replace “pinpoit” with “pinpoint”. A. Thanks, corrected

  1. In line 384, check spelling: replace “life-treathening” with “life-threatening”. A. Thanks, corrected

  1. In line 387, check spelling: replace “understaind” with either “understand” or

“understanding” (not sure which was intended). A. thanks, corrected “understanding”

  1. Please check for consistency of spelling (American English or British English). If using

American English:

  1. “neighbouring” (lines 111, 116) should be spelt as “neighboring”;
  2. “hyperglycaemia (line 134) should be spelt as “hyperglycemia”;
  3. “haemorrhage” (line 293) should be spelt as “hemorrhage”.

A: thanks, 2 a-c corrected as suggested

  1. Please check for proper hyphenation and spacing:
  2. In line 82, check hyphenation: replace “blood borne” with “blood-borne”.
  3. In line 85, check spacing: “artery,to” – add space between the comma and “to”.
  4. In line 116, check hyphenation: replace “tissue specific” with “tissue-specific”.
  5. In line 206, check hyphenation: replace “corticosteroid based therapy” with

“corticosteroid-based therapy”.

  1. In line 349, check spacing: remove extra space after hyphen in “marker- based”.
  2. In lines 354, 383 and 385, check hyphenation: replace “diabetes associated” with

“diabetes-associated”.

  1. In line 400 (Figure 6 caption), check spacing: “deliveryare”.
  2. In line 411, check hyphenation: replace “neurotrophins related” with “neurotrophinrelated”.
  3. In lines 421 to 422, check hyphenation: replace “well established” with “wellestablished”.
  4. In line 425, check spacing: delete space after hyphen “streptozotocin (STZ)–

induced”.

A: A: thanks, 3 a-j corrected as suggested

  1. Please use abbreviations appropriately and spell out all abbreviations in full form at first use:
  2. In the abstract, if the abbreviation “DR” is used (in lines 23 and 24), it should be

defined in parentheses at first use (line 18).

  1. In line 38, “miRNA signatures (miRNA)” – miRNA is not an abbreviation of miRNA.

First use should be “microRNA”.

  1. In line 55, the full form of “SGLT2i” should be given first followed by the

abbreviation “SGLT2i” in parentheses. In the following lines (lines 60 and 62), a

different abbreviation “SGLT-2i” is used. Please check for consistency.

  1. In line 58, please spell out “DPP-4i” in full form at first use.
  2. In line 141 (Figure 2 caption), please define abbreviations “RAGE” and “ROS” (even if

already defined in the main text).

  1. In line 218, please spell out “HFpEF” in full form at first use.
  2. In line 231, please spell out “DCM” in full form at first use.
  3. In line 282, please spell out “HFrEF” in full form at first use.
  4. In line 364, please spell out “STZ” in full form at first use.
  5. In line 471, please spell out “API” in full form at first use.
  6. In line 478, please spell out “GDNF” and “FGF-2” in full form at first use.
  7. In line 479, please spell out “CNTF” and “OSM” in full form at first use.
  8. In line 482, please spell out “PEDF” in full form at first use.

A: thanks, 4a-m corrected as suggested

  1. Please avoid using abbreviations in the headings:
  2. In line 130, replace “iBRB” with “internal blood-retinal barrier”.
  3. In line 470, replace “NT” with “neurotrophin”.

A: thanks,5a-b corrected as suggested

  1. For greater clarity, it is recommended to avoid using abbreviations unnecessarily, especially

abbreviations that appear only once.

A: thanks, we realized this important point.

Others:

  1. In line 30, Section 1 heading: is “state-of-the-art” commonly used this way? Suggest just

using the heading “Introduction”.

  1. thanks, corrected “introduction”

  1. In line 39, suggest to replace “might have the potential serving as” with “may potentially

serve as”.

  1. thanks, corrected as suggested.
  2.  
  1. In line 41, “DM has several grades, including type 1, type 2, …” In this context, “grades” may

not be the most appropriate term, as clinical grading usually refers to the severity of disease

(mild, moderate, severe, etc). An alternative term, such as “types” or “classifications”, may

be more suitable.

  1. thanks, corrected “types”

  1. In line 43, “(Zitat?)” may have been left in the manuscript by accident.
  2. yes, thanks, deleted.

  1. In lines 123 to 124, replace “Not only sharing…” with “Not only do they share…”
  2. thanks, corrected “Not only do they share“

  1. In line 126, “Accordingly, we recently showed …” Did you show or did the authors of that

study show?

  1. In line 127, check placement of commas: suggest to place first comma after “affects”, i.e., “…

entire eye in mice affects, amongst other factors, pericytes and endothelial cells”.

  1. In line 133, replace “demonstrate at certain symptoms of DR” with “demonstrate certain

symptoms of DR”. (Or did you mean “demonstrate, at certain times, symptoms of DR”?)

  1. In lines 139 to 140, not sure what is “comp. [163]”.

  1. In line 144, remove the letter “s” in front of “Figure 3”.
  2. thanks, removed.
  3. In line 172, replace “diabetic associated macula edema” with “diabetic macular edema”.

Please note “macular” with the “r”.

  1. thanks, rephrased acc. Reviewer 1
  2.  
  1. In line 206, suggest replacing “diabetic associated macular edema” with “diabetic macular

edema”, which is the official medical term.

  1. thanks, rephrased acc. Reviewer 1

  1. In line 208, replace “not being limited to” with “are not limited to”.
  2. thanks replaced

  1. In line 233, suggest to delete “such as”.
  2. thanks, deleted

  1. In line 237, delete “Role of microangiopathy in diabetic cardiomyopathy” (title of article?) –

should not be part of the sentence.  

  1. thanks, deleted

  1. In line 262, remove “(Deckert T, Yokoyama H, Mathiesen E, et al.”
  2. thanks, removed

  1. In line 281, replace “the signatures of microRNA signatures (miRNA)” with either “the

signatures of microRNA (miRNA)” or “the microRNA (miRNA) signatures”.

  1. thanks was replaced “the microRNA (miRNA) signatures”

  1. In line 295, replace “Early trials show raise confidence” with either “Early trials show raised

confidence” or “Early trials raise confidence”.

  1. thanks, replaced: Early trials raise confidence
  2.  
  1. In line 324, replace colon with full stop.
  2. thanks, replaced

  1. In line 328, “could be shown” or “has been shown”?
  2. thanks, replaced

  1. In line 332, remove extra full stop.
  2. thanks, removed

  1. In line 348, recommend changing “state” to “stage”.
  2. thanks replaced

  1. Suggest using “microRNA” or the abbreviated form “miRNA” consistently, instead of

alternating between the two.

  1. thanks, decided for miRNA

  1. In line 352, replace “is very promising avenue” with “is a very promising avenue”.
  2. thanks replaced

  1. In line 366, remove the letter “s” before “Palmitate”.
  2. thanks, removed

  1. In line 370, not sure what is “Remarkaaus morbly”.
  2. thanks, re-edited

  1. In line 380, replace “perspective” with “prospective”.
  2. thanks replaced

  1. In line 382, replace “By” with “by” (lowercase “b”).
  2. thanks replaced

  1. In line 384, replace “to tackling” with “to tackle”.
  2. thanks replaced

  1. In line 386, replace “utilizing the miRNAtherapeutics” with “utilizing miRNA therapeutics”.
  2. thanks replaced

  1. In line 394 (Figure 5 caption), replace “miRNA inhibitors of miRNAs” with either “miRNA

inhibitors” or “inhibitors of miRNAs”.

  • thanks replaced “miRNA inhibitors”
  1.  
  1. In line 404, replace “has been” with “have been”.
  2. thanks replaced

  1. In line 436, citation [135] should be placed outside the parentheses.
  2. thanks replaced

  1. In line 477, replace “hrs” with “hours”.
  2. thanks replaced

  1. In line 484, please cite “(Di Pierdomenico et al., 2018)” in the appropriate format consistent

with the rest of the article.

  1. In line 489, please cite “(McGill et al., 2007; Bok et al., 2002)” in the appropriate format

consistent with the rest of the article.

  1. Sorry, was leftover internal information. Was deleted.

Reviewer 3 Report

This review by Sergey Shityakov et al. describes the impact of diabetes mellitus on the retina, heart, and kidneys, describing a common putative microRNA signature in diabetic retinopathy, diabetic nephropathy, and heart failure, which could potentially be used as a biomarker in the future to better monitor disease progression. 

The review contributes to the field, but there are major Concerns that should be improved to make it well organized and comprehensively described,: 

1) The authors should better describe the role of anti-glycaemic drugs in the prevention and or treatment of Diabetic retinopathy, Diabetic nephropathy and Heart failure. A recent meta-analysis (PMID: 34418562), showed that the pooled intention-to-treat analysis showed a reduced risk of stroke with SGLT2 inhibitors compared to DPP-4 inhibitors (Hazard ratio HR, 0.89; 95%CI, 0.82-0.96; I2 = 25%; p = 0.25) and non-SGLT2 inhibitors (HR, 0.83; 95%CI, 0.77-0.91; I2 = 11%; p = 0.34). Finally, SGLT2 inhibitors were also associated with reduced CV outcomes and mortality risk in all comparisons. 

2) miR-21 is the most studied star microRNA in diabetes and cardiomyopathies. In the part 4.2, its paradoxical role in diabetes and diabetes-related heart diseases should be discussed (PMID: 34778418).

Author Response

This review by Sergey Shityakov et al. describes the impact of diabetes mellitus on the retina, heart, and kidneys, describing a common putative microRNA signature in diabetic retinopathy, diabetic nephropathy, and heart failure, which could potentially be used as a biomarker in the future to better monitor disease progression. 

The review contributes to the field, but there are major Concerns that should be improved to make it well organized and comprehensively described,: 

  1. The authors should better describe the role of anti-glycaemic drugs in the prevention and or treatment of Diabetic retinopathy, Diabetic nephropathy and Heart failure. A recent meta-analysis (PMID: 34418562), showed that the pooled intention-to-treat analysis showed a reduced risk of stroke with SGLT2 inhibitors compared to DPP-4 inhibitors (Hazard ratio HR, 0.89; 95%CI, 0.82-0.96; I2 = 25%; p = 0.25) and non-SGLT2 inhibitors (HR, 0.83; 95%CI, 0.77-0.91; I2 = 11%; p = 0.34). Finally, SGLT2 inhibitors were also associated with reduced CV outcomes and mortality risk in all comparisons. 

2) miR-21 is the most studied star microRNA in diabetes and cardiomyopathies. In the part 4.2, its paradoxical role in diabetes and diabetes-related heart diseases should be discussed (PMID: 34778418).

Another promising biomarker to diagnose various cardiomyopathies, including heart failure and diabetic cardiomyopathy, might be miR-21 (Surina et al., 2021). This 22-nucleotide long microRNA has a paradoxical effect on cardiomyocytes against stress overloaded conditions. It activates fibroblast to trigger the fibrosis process stimulating the proliferation of the heart cells at the same time (Surina et al., 2021) Besides, the high level of miR-21 expression might be linked to the suppression of programmed cell death protein 4 in head and neck squamous cell carcinoma (Ajuyah et al., 2019).

Surina, S., et al. (2021). "miR-21 in Human Cardiomyopathies." Front Cardiovasc Med 8: 767064.

Round 2

Reviewer 1 Report

Now the authors manage to address all the major concerns. Consequently, the quality of manuscript is significantly increased. I have only two minor concerns

Minor point:

1) The formatting of the revised manuscript at certain pages does not seem to keep the margins. Please align margins appropriately. But this might be arranged by the production division of the publisher.

2) Row 132: "2. Eyes, renal and cardiac microvasculature - ...". Instead of this formulation I suggest Retinal (or ocular), renal and cardiac... given that eyes is not an adjective form.

Author Response

Thanks

Reviewer 3 Report

The authors answered  all concerns, and the manuscript can be accepted in present form

Author Response

Thanks